JCB Journal of Cell Biology

# Kinetoplastid kinetochore proteins KKT2 and KKT3 have unique centromere localization domains

Gabriele Marcianò*, Midori Ishii*, Olga O. Nerusheva*, and Bungo Akiyoshi

**The kinetochore is the macromolecular protein complex that assembles onto centromeric DNA and binds spindle microtubules. Evolutionarily divergent kinetoplastids have an unconventional set of kinetochore proteins. It remains unknown how kinetochores assemble at centromeres in these organisms. Here, we characterize KKT2 and KKT3 in the kinetoplastid parasite *Trypanosoma brucei*. In addition to the N-terminal kinase domain and C-terminal divergent polo boxes, these proteins have a central domain of unknown function. We show that KKT2 and KKT3 are important for the localization of several kinetochore proteins and that their central domains are sufficient for centromere localization. Crystal structures of the KKT2 central domain from two divergent kinetoplastids reveal a unique zinc-binding domain (termed the CL domain for centromere localization), which promotes its kinetochore localization in *T. brucei*. Mutations in the equivalent domain in KKT3 abolish its kinetochore localization and function. Our work shows that the unique central domains play a critical role in mediating the centromere localization of KKT2 and KKT3.**

## Introduction

The kinetochore is the macromolecular protein complex that drives chromosome segregation during mitosis and meiosis in eukaryotes. Its fundamental functions are to bind DNA and spindle microtubules (Musacchio and Desai, 2017). In most eukaryotes, kinetochores assemble within a single chromosomal region called the centromere. While components of spindle microtubules are highly conserved across eukaryotes (Wickstead and Gull, 2011; Findeisen et al., 2014), centromere DNA is known to evolve rapidly (Henikoff et al., 2001). Nevertheless, it is critical that a single kinetochore is assembled per chromosome and its position is maintained between successive cell divisions. A key player involved in this kinetochore specification process is the centromere-specific histone H3 variant, CENP-A, which is found in most sequenced eukaryotic genomes (Talbert et al., 2009). CENP-A localizes specifically at centromeres throughout the cell cycle and recruits HJURP, a specific chaperone that incorporates CENP-A onto centromeres (Black and Cleveland, 2011; McKinley and Cheeseman, 2016; Stankovic and Jansen, 2017). Besides CENP-A, components of the constitutive centromere-associated network also localize at centromeres throughout the cell cycle. CENP-A–containing nucleosomes are recognized by constitutive centromere-associated network components, which in turn recruit the KNL1-Mis12-Ndc80 network, which has microtubule-binding activities. In addition to these structural kinetochore proteins, several protein kinases are known to localize at mitotic kinetochores, including Cdk1, Aurora B, Bub1,

Mps1, and Plk1 (Cheeseman and Desai, 2008). These protein kinases regulate various aspects of mitosis, including kinetochore assembly, error correction, and the spindle checkpoint (Carmena et al., 2012; London and Biggins, 2014; Hara and Fukagawa, 2018).

Kinetoplastids are evolutionarily divergent eukaryotes that are defined by the presence of a unique organelle called the kinetoplast, which contains a cluster of mitochondrial DNA (d'Avila-Levy et al., 2015). Centromere positions have been mapped in three kinetoplastids: 20–120-kb regions that have AT-rich repetitive sequences in *Trypanosoma brucei* (Obado et al., 2007; Echeverry et al., 2012), ~16-kb GC-rich unique sequences in *Trypanosoma cruzi* (Obado et al., 2005), and ~4-kb regions in *Leishmania major* (Garcia-Silva et al., 2017). Although some DNA elements and motifs are enriched, there is no specific DNA sequence that is common to all centromeres in each organism, suggesting that kinetoplastids likely determine their kinetochore positions in a largely sequence-independent manner; however, none of CENP-A or any other canonical structural kinetochore protein has been identified in kinetoplastids (Lowell and Cross, 2004; Berriman et al., 2005; Aslett et al., 2010). They instead have unique kinetochore proteins, such as KKT1–25 (Akiyoshi and Gull, 2014; Nerusheva and Akiyoshi, 2016; Nerusheva et al., 2019) and KKIP1–12 (D'Archivio and Wickstead, 2017; Brusini et al., 2021) in *T. brucei*. Some of these kinetochore proteins have similarities to meiotic synaptonemal complex or

Department of Biochemistry, University of Oxford, Oxford, UK.

*G. Marcianò, M. Ishii, and O.O. Nerusheva contributed equally to this paper; Correspondence to Bungo Akiyoshi: bungo.akiyoshi@bioch.ox.ac.uk.



homologous recombination components, suggesting that kinetoplastids might have evolved their unique kinetochore system by repurposing components of the chromosome synapsis and homologous recombination machinery (Tromer et al., 2021). It remains unknown which kinetochore proteins form the base of kinetoplastid kinetochores that recruits other proteins. There are six proteins that localize at centromeres throughout the cell cycle (KKT2, KKT3, KKT4, KKT20, KKT22, and KKT23), implying their close association with centromeric DNA. Indeed, we previously showed that KKT4 has DNA-binding activity in addition to microtubule-binding activity (Llauró et al., 2018; Ludzia et al., 2021); however, RNAi-mediated knockdown of KKT4 affected the localization of KKT20, but not other kinetochore proteins, suggesting that KKT4 is largely dispensable for kinetochore assembly.

In this study, we focused on KKT2 and KKT3, which are homologous to each other and have three domains conserved among kinetoplastids: A protein kinase domain classified as unique among known eukaryotic kinase subfamilies (Parsons et al., 2005), a central domain of unknown function, and divergent polo boxes. The presence of an N-terminal kinase domain and C-terminal divergent polo boxes suggests that KKT2 and KKT3 likely share common ancestry with polo-like kinases (Nerusheva and Akiyoshi, 2016). Interestingly, a protein kinase domain is not present in any constitutively localized kinetochore protein in other eukaryotes, highlighting these protein kinases as a unique feature of kinetoplastid kinetochores. In addition to the three domains that are highly conserved among kinetoplastids, AT-hook and SPKK DNA-binding motifs are found in some species, suggesting that these proteins are located close to DNA (Akiyoshi and Gull, 2014). Although RNAi-mediated knockdown of KKT2 or KKT3 leads to growth defects (Akiyoshi and Gull, 2014; Jones et al., 2014), little is known about their molecular function. In this report, we have revealed a unique zinc-binding domain in the KKT2 and KKT3 central domain, which is important for their kinetochore localization and function in *T. brucei*.

## Results

### Localization of KKT2 and KKT3 is not affected by depletion of various kinetochore proteins

We previously showed in *T. brucei* procyclic form (insect stage) cells that kinetochore localization of KKT2 and KKT3 was not affected by KKT4 depletion (Llauró et al., 2018). To examine the effect of other kinetochore proteins for the recruitment of KKT2 and KKT3, we established RNAi-mediated knockdowns for KKT1, KKT6, KKT7, KKT8, KKT10/19, KKT14, KKT22, KKT23, KKT24, and KKIP1 (Fig. 1 A). Using inducible stem-loop RNAi constructs in cells expressing YFP fusion of the corresponding target protein, we confirmed efficient depletion of YFP signals and observed severe growth defects for KKT1, KKT6, KKT8, KKT14, and KKT24. RNAi against KKT7, KKT10/19, and KKIP1 also caused severe growth defects, as previously observed (D'Archivio and Wickstead, 2017; Ishii and Akiyoshi, 2020). In contrast, a mild growth defect was observed for KKT23, while no growth defect was observed for KKT22. We next used these RNAi constructs in cells expressing either YFP-KKT2 or KKT3-

YFP and found that these proteins formed kinetochore-like dots in all conditions (Fig. 1, B and C). These results show that KKT2 and KKT3 can localize at kinetochores even when various kinetochore proteins are depleted.

### KKT2 and KKT3 are important for localization of some kinetochore proteins

We next examined whether KKT2 and KKT3 are important for the localization of other kinetochore proteins. KKT2 RNAi using a stem-loop construct caused growth defects, as previously reported (Fig. 2 A; Ishii and Akiyoshi, 2020). We saw defective kinetochore localization for KKT14 upon induction of KKT2 RNAi, while other tested proteins still formed kinetochore-like dots at 1 d after induction (Fig. 2, B and C). We next established RNAi against KKT3 (Fig. 2 D) and examined its effect on the localization of other kinetochore proteins at 2 d after induction. Kinetochore localization of two constitutive kinetochore components, KKT22 and KKT23, was affected by KKT3 depletion (Fig. 2, E and F), while that of other tested proteins remained largely intact. It is noteworthy that KKT2 and KKT3 can apparently localize at kinetochores independently from each other. Together with the fact that KKT2 and KKT3 are homologous proteins, these results raised the possibility that KKT2/3 might have redundant roles in kinetochore assembly, so we next examined the effect of double knockdown. The growth defect of KKT2/3 double RNAi was not dramatically different from that of individual knockdowns (Fig. 2 G). We also found that the percentage of cells that had defective kinetochore localization of KKT14 was lower in KKT2/3 double RNAi (22%; Fig. 2 H) compared with KKT2 RNAi (46%; Fig. 2 B). These results suggest that depletion of KKT2 proteins in KKT2/3 double RNAi was not as efficient as that in KKT2 single RNAi, which is consistent with the residual KKT2 signals observed in KKT2/3 double-RNAi cells (Fig. 2 I). Despite this limitation, defective kinetochore localization was found for KKT1 and KKT4 at 1 d after induction (Fig. 2, H and I), suggesting that their kinetochore localization depends on both KKT2 and KKT3. It is possible that more efficient or rapid inactivation methods could reveal additional kinetochore proteins whose localizations depend on KKT2/3. Taken together, these results show that KKT2 and KKT3 play important roles in recruiting multiple kinetochore proteins.

### Multiple domains of KKT2 are able to localize at centromeres in *T. brucei*

To understand how KKT2 localizes at centromeres, we determined which domain was responsible for its centromere localization by expressing a series of truncated versions of KKT2, fused with a GFP-tagged nuclear localization signal (NLS) peptide (GFP-NLS) in *T. brucei* (Fig. 3 A). We previously showed that an ectopically expressed KKT2 divergent polo box (DPB) domain (residues 1,024–1,260) localized at kinetochores (Nerusheva and Akiyoshi, 2016). The present study confirmed this result and also identified two other regions (residues 562–677 and 672–1,030) that localized at kinetochores from S phase to anaphase (Fig. 3, A and B; and Fig. S1).

Based on our previous finding that KKT2 coimmunoprecipitated with a number of kinetochore proteins (Akiyoshi and Gull, 2014),

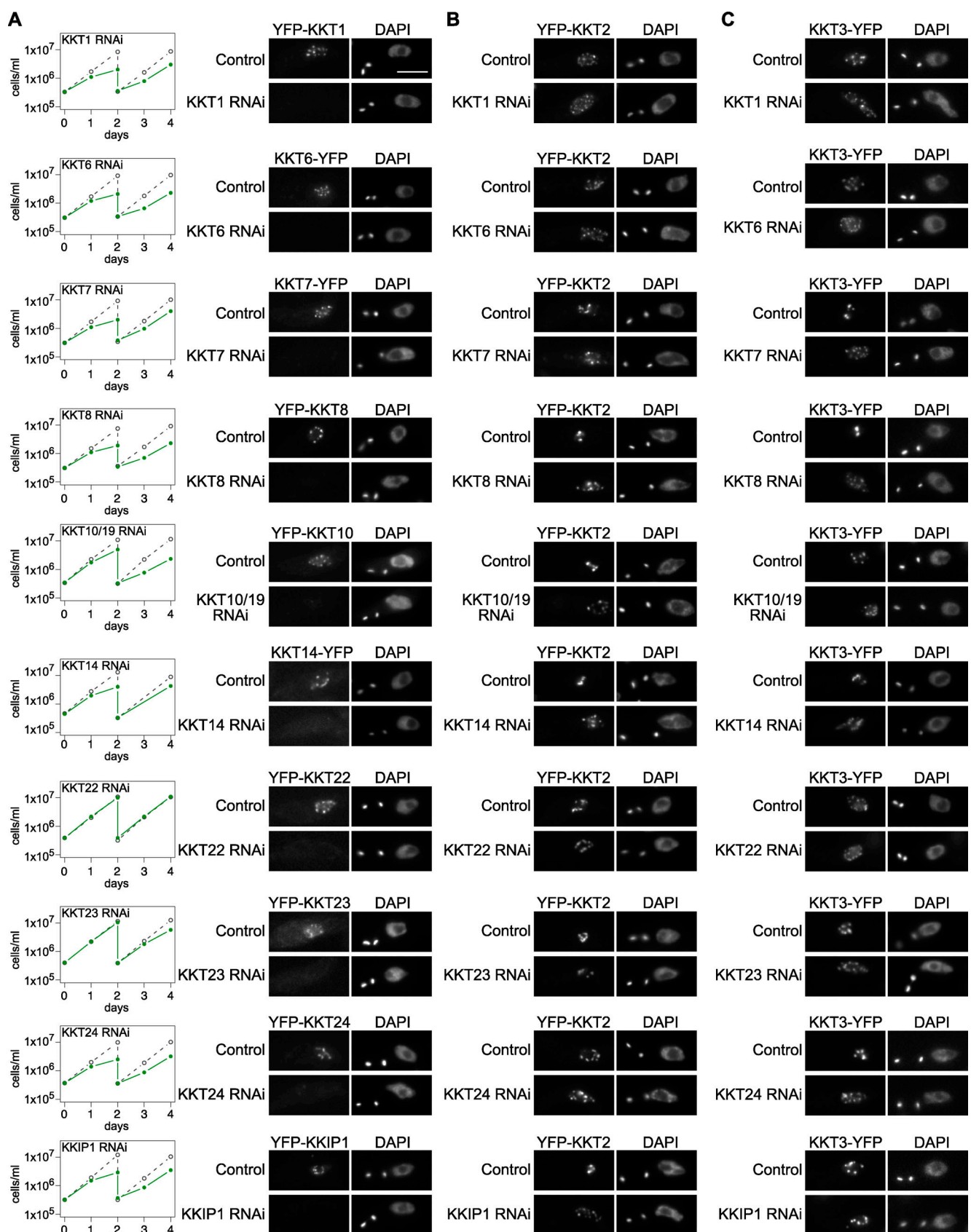

Figure 1.   **Kinetochore localization of KKT2 and KKT3 is not dependent on various kinetochore proteins. (A)** Left: Growth curves of control (gray dashed line) and RNAi-induced cultures (green line) show that these kinetochore proteins are important for proper cell growth (except KKT22) in *T. brucei* procyclic

form cells. RNAi was induced with 1 µg/ml doxycycline. Cultures were diluted at day 2. Right: Depletion of indicated kinetochore proteins was confirmed by microscopy. Cells were fixed at either day 1 (RNAi against KKT1, KKT6, KKT7 5′UTR, KKT8 3′UTR, KKT10/19, KKT14, KKT24, and KKIP1) or day 2 (RNAi against KKT22 and KKT23). *T. brucei* has a kinetoplast (K) that contains mitochondrial DNA and a nucleus (N) that contains nuclear DNA. Kinetoplasts segregate before the nuclear division, and the number of kinetoplasts and nuclei can be used as a cell cycle marker (Woodward and Gull, 1990; Siegel et al., 2008). YFP signal was efficiently depleted in >60% of 2K1N cells (G2 to metaphase) in each case (n > 50 each). Cell lines: BAP672, BAP699, BAP2001, BAP2002, BAP139, BAP2086, BAP1840, BAP1842, BAP1843, and BAP770. **(B)** Kinetochore localization of KKT2 is not affected by depletion of various kinetochore proteins. RNAi against indicated kinetochore proteins was induced in cells expressing YFP-KKT2. Dot formation was observed in >94% of 2K1N cells in all cases (n > 70 each). Cells were fixed as in A, except for KKT23 RNAi cells that were fixed at day 4. Cell lines: BAP2004, BAP2005, BAP2006, BAP2007, BAP2008, BAP680, BAP2010, BAP2011, BAP2012, and BAP2013. **(C)** Kinetochore localization of KKT3 is not affected by depletion of various kinetochore proteins. RNAi against indicated kinetochore proteins was induced in cells expressing KKT3-YFP. Dot formation was observed in >97% of 2K1N cells in all cases (n > 50 each). Cells were fixed as in B. Cell lines: BAP2014, BAP2015, BAP2016, BAP2017, BAP2018, BAP2085, BAP2020, BAP2021, BAP2022, and BAP2023. Scale bar, 5 µm.

we reasoned that these fragments might localize at kinetochores by interacting with other kinetochore proteins. To test this possibility, we immunoprecipitated KKT2 fragments and performed mass spectrometry to identify copurifying proteins. Although the central domain copurified only with limited amounts of KKT3 and KKT5 (Fig. 3 C and Table S1), KKT2$^{672–1030}$ copurified with several kinetochore proteins, with KKIP1 being the top hit (Fig. 3 D and Table S1). Similarly, KKT2 DPB copurified with several kinetochore proteins, which was abolished in the W1048A mutant that did not localize at kinetochores (Fig. 3 E and Table S1; Nerusheva and Akiyoshi, 2016). These results support the possibility that ectopically expressed KKT2$^{672–1030}$ and DPB are able to localize at kinetochores from S phase to anaphase by interacting with nonconstitutive kinetochore proteins (e.g., KKT1, KKT6, KKT7, KKT8, and KKIP1). A corollary is that, in WT cells, the constitutively localized KKT2 protein recruits these transient kinetochore proteins onto kinetochores using KKT2$^{672–1030}$ and DPB domains. It is noteworthy that KKT14 was not detected in the immunoprecipitates of any KKT2 fragments we tested, despite our findings that KKT2 was one of the most abundant proteins in the immunoprecipitates of KKT14 (Akiyoshi and Gull, 2014) and that kinetochore localization of KKT14 depends on KKT2 (Fig. 2 B). It is possible that KKT14 copurifies with a different KKT2 fragment.

## The central domain of KKT3 is able to localize at centromeres constitutively

We next expressed KKT3 fragments in trypanosomes (Fig. 4, A and B). Similar to KKT2, the N-terminal protein kinase domain of KKT3 did not localize at centromeres. Although KKT2 DPB and KKT3 DPB are 25% identical in primary sequence (Akiyoshi and Gull, 2014), KKT3 DPB had robust kinetochore localization only during anaphase (Fig. 4 C), which differs from KKT2 DPB that localized from S phase to anaphase. Immunoprecipitation of KKT3 DPB identified a number of copurifying kinetochore proteins (Fig. 4 D), raising a possibility that KKT3, like KKT2, recruits other kinetochore proteins by its DPB.

KKT3$^{594–1058}$ and KKT3$^{594–811}$ that contain the central region also localized at kinetochores. Kinetochore localization was also observed for KKT3$^{594–728}$, which lacks AT-hook and SPKK DNA-binding motifs (residues 771–780). KKT3$^{594–811}$ copurified with limited amounts of KKT7, KKT1, and KKT2 (Fig. 4 E). Importantly, KKT3$^{594–728}$ constitutively localized at kinetochores (Fig. S1), suggesting that the central domain plays a crucial role in recruiting KKT3 onto centromeres throughout the cell cycle.

## The *Bodo saltans* KKT2 central domain adopts a unique structure

To gain insight into how the central domains of KKT2 and KKT3 localize at centromeres, we expressed and purified recombinant proteins for their structure determination by x-ray crystallography. Our attempts to purify the *T. brucei* KKT3—referred to TbKKT3 hereafter—central domain were unsuccessful, but we managed to express and purify from *Escherichia coli* the central domain of KKT2 from several kinetoplastids, including *Bodo saltans* (a free-living kinetoplastid; Jackson et al., 2016) and *Perkinsela* sp. (endosymbiotic kinetoplastids; Fig. S2; Tanifuji et al., 2017). We obtained crystals of BsKKT2$^{572–668}$, which corresponds to residues 569–664 in *T. brucei* KKT2, and determined its structure to 1.8-Å resolution by zinc single-wavelength anomalous dispersion (Zn-SAD) phasing (Fig. 5 and Table 1). Our analysis revealed the presence of two distinct zinc-binding domains: The N-terminal one (referred to as the CL domain for its key role in centromere localization; see below) consists of two β-sheets (where β-strands 1, 4, and 5 comprise the first β-sheet, and β-strands 2 and 3 comprise the second β-sheet) and one α-helix, while the C-terminal one consists of one β-sheet (comprising β-strands 6 and 7) and one α-helix (Fig. 5, A and B). The CL domain coordinates two zinc ions and the C-terminal domain coordinates one zinc ion.

A structural homology search using the DALI server (Holm and Laakso, 2016) indicated that the CL domain has weak structural similarity to proteins that have C1 domains (Table S2). C1 domains were originally discovered as lipid-binding modules in PKCs and are characterized by the HX$_{12}$CX$_2$CXnCX$_2$CX$_4$HX$_2$CX$_7$C motif (Colón-González and Kazanietz, 2006; Das and Rahman, 2014). C1 domains are classified into a typical C1 domain that binds diacylglycerol or phorbol esters, and an atypical C1 domain not known to bind ligands. The closest structural homologue of the BsKKT2 CL domain was the atypical C1 domain of the Vav1 protein (root mean square deviation [RMSD]: 2.7 Å across 52 Cα). Although the CL domain and the C1 domain share some structural similarity, their superposition revealed fundamental differences (Fig. 6). Coordination of one zinc ion in the BsKKT2 CL domain occurs via the N-terminal residues Cys580 and His584, while that in the Vav1 C1 domain occurs via the N-terminal His516 and C-terminal Cys564. More importantly, the CL domain does not have the HX$_{12}$CX$_2$CXnCX$_2$CX$_4$HX$_2$CX$_7$C motif that is present in all C1 domains. Therefore, the structural similarity of these two distinct domains is likely a product of convergent evolution.

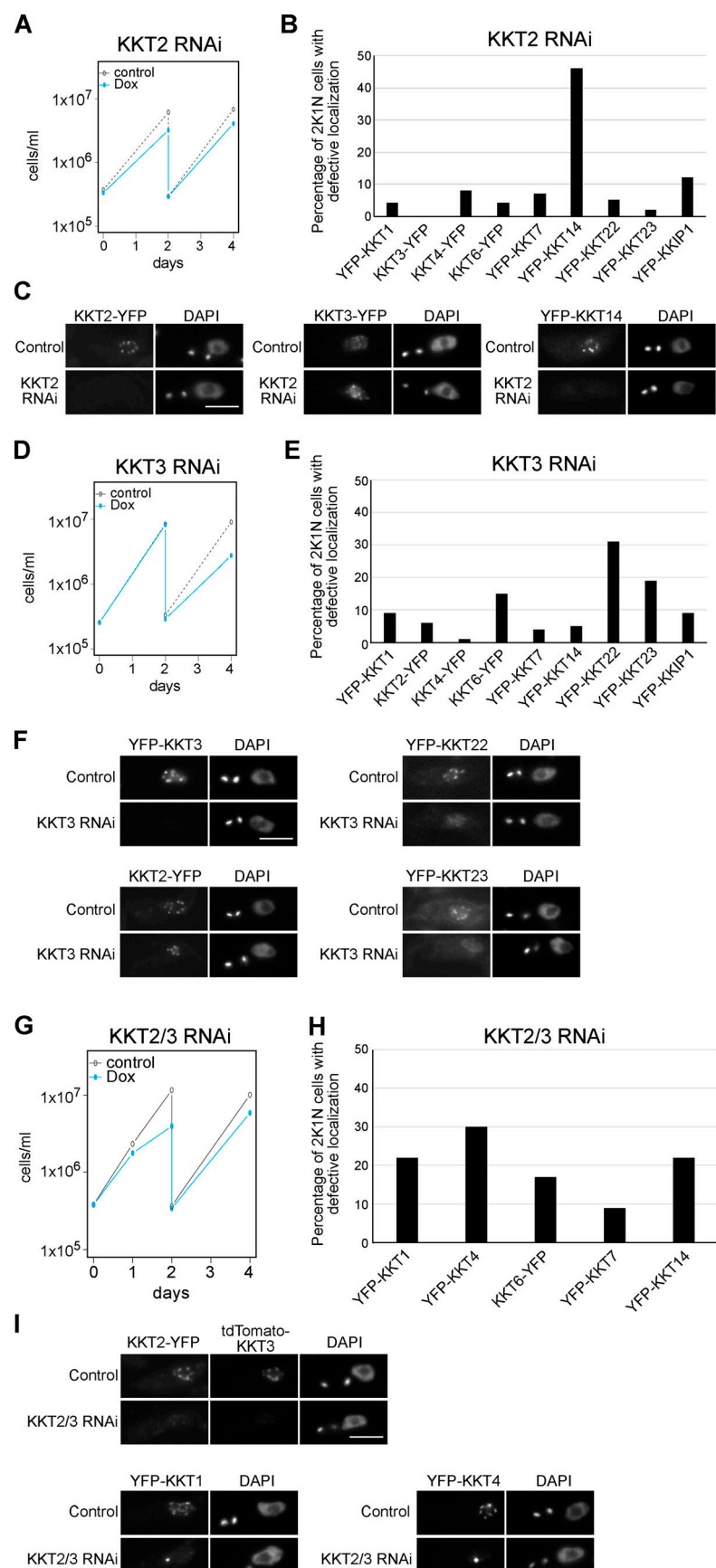

Figure 2. **KKT2 and KKT3 are important for kinetochore assembly. (A)** Growth curve for KKT2 5′UTR RNAi in cells expressing KKT2-YFP. RNAi was induced with 1 µg/ml doxy-cycline (Dox) and cultures were diluted at day 2. KKT2-YFP signal was depleted in 46% of 2K1N cells (G2 to metaphase) at day 1 after induction (n > 100). Cell line: BAP1752. **(B)** Quantification of 2K1N cells that had defective kinetochore localization of indicated kinetochore proteins upon induction of KKT2 5′UTR RNAi (n > 100 each). Cells were fixed at day 1 after induction. In each case, defective localization of YFP signal was found in <6% of uninduced 2K1N cells (not shown). Cell lines: BAP1743, BAP1753, BAP1749, BAP2075, BAP1750, BAP1746, BAP1751, BAP2080, and BAP1747. **(C)** Examples of cells expressing indicated kinetochore proteins fused with YFP, showing that KKT3 still forms kinetochore-like dots, while KKT14 fails to localize at kinetochores upon KKT2 depletion. **(D)** Growth curve for KKT3 3′UTR RNAi in cells expressing YFP-KKT3. Cultures were diluted at day 2. YFP-KKT3 signal was depleted in 61% of 2K1N cells at day 2 after induction (n > 100). Cell line: BAP1659. **(E)** Quantification of 2K1N cells that had defective kinetochore localization of indicated kinetochore proteins upon induction of KKT3 3′UTR RNAi for 2 d (n > 100, each). In each case, defective localization of YFP signal was found in <3% of uninduced 2K1N cells (not shown). Cell lines: BAP1755, BAP1764, BAP1761, BAP2076, BAP1762, BAP1758, BAP1763, BAP2081, and BAP1759. **(F)** Examples of cells expressing indicated kinetochore proteins fused with YFP, showing that KKT2 still forms kinetochore-like dots, while KKT22 and KKT23 failed to localize at kinetochores upon KKT3 depletion. **(G)** Growth curve for KKT2/3 double RNAi that targets KKT2 5′UTR and KKT3 3′UTR in cells expressing KKT2-YFP and tdTomato-KKT3. Both KKT2 and KKT3 signals were depleted in 55% of 2K1N cells at day 1 after induction (n > 100). Cell line: BAP2089. **(H)** Quantification of 2K1N cells that had defective kinetochore localization of indicated kinetochore proteins upon induction of KKT2/3 double RNAi for 1 d (n > 100 each). In each case, defective localization of YFP signal was found in <6% of uninduced 2K1N cells (not shown). Cell lines: BAP2068, BAP2070, BAP2074, BAP2071, and BAP2073. **(I)** Examples of cells expressing indicated kinetochore proteins fused with fluorescent proteins, showing that KKT1 and KKT4 failed to form normal kinetochore-like dots but instead formed bright blobs upon KKT2/3 depletion. Cells were fixed at day 1 after induction. Scale bars, 5 µm.

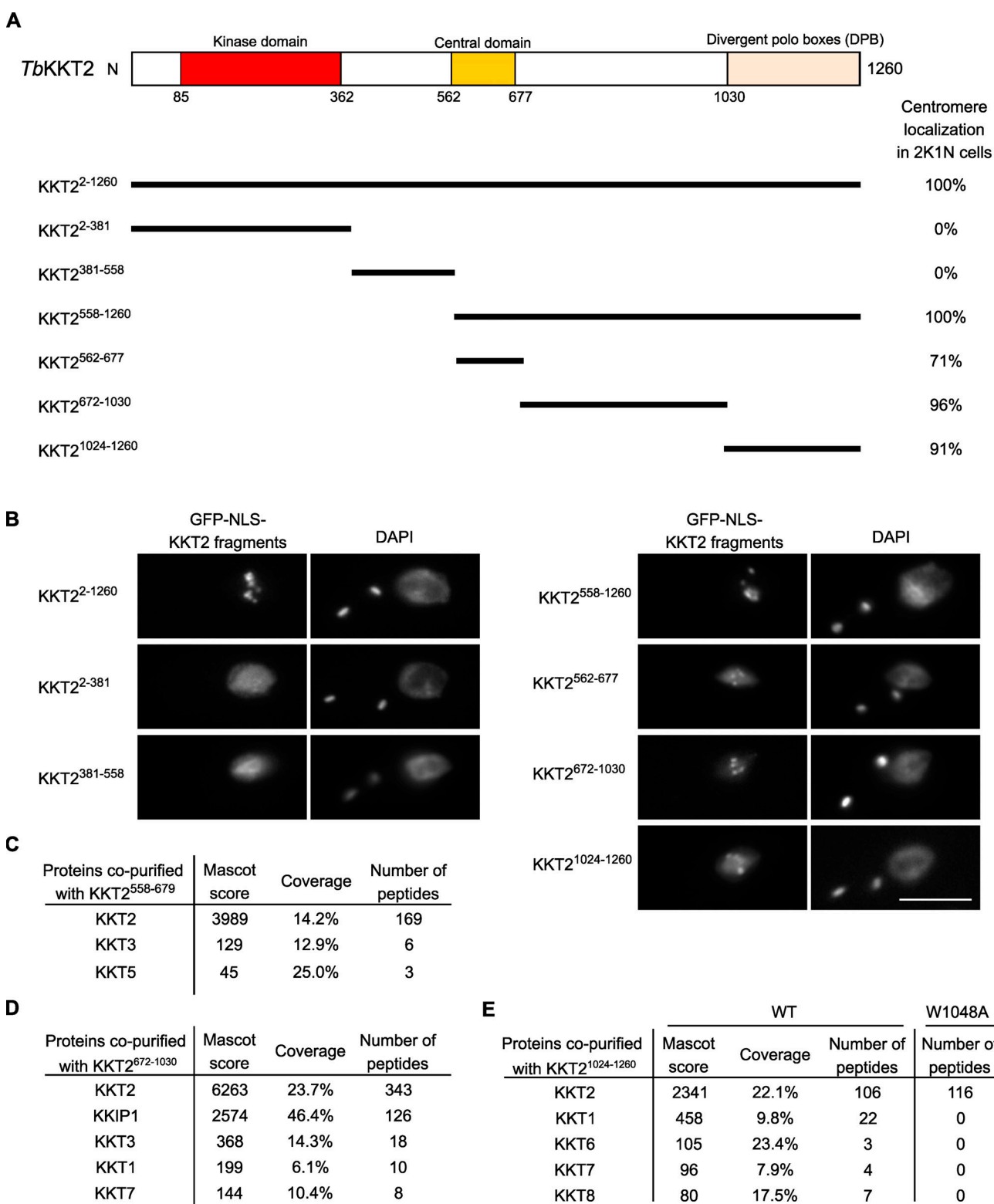

Figure 3. **KKT2 has multiple domains that can promote centromere localization in *T. brucei*. (A)** Schematic of the *T. brucei* KKT2 protein. Percentages of GFP-positive 2K1N cells (G2 to metaphase) that have kinetochore-like dots were quantified at 1 d after induction (*n* > 22 each). **(B)** Ectopically expressed *Tb*KKT2 fragments that contain either the central domain (562–677), 672–1,030, or the divergent polo boxes (1,024–1,260) form kinetochore-like dots. Inducible GFP-NLS fusion proteins were expressed with 10 ng/ml doxycycline. Cell lines: BAP327, BAP328, BAP381, BAP331, BAP457, BAP519, and BAP517. Scale bar, 5 µm. **(C)** *Tb*KKT2⁵⁵⁸⁻⁶⁷⁹ does not copurify robustly with other kinetochore proteins. Cell line: BAP382. **(D)** *Tb*KKT2⁶⁷²⁻¹⁰³⁰ copurifies with KKIP1 and several other kinetochore proteins. Cell line: BAP519. **(E)** *Tb*KKT2 DPB WT, not W1048A, copurifies with several kinetochore proteins. Cell lines: BAP517 and BAP535. Inducible GFP-NLS fusion proteins were expressed with 10 ng/ml doxycycline, and immunoprecipitation was performed using anti-GFP antibodies. See Table S1 for all proteins identified by mass spectrometry.

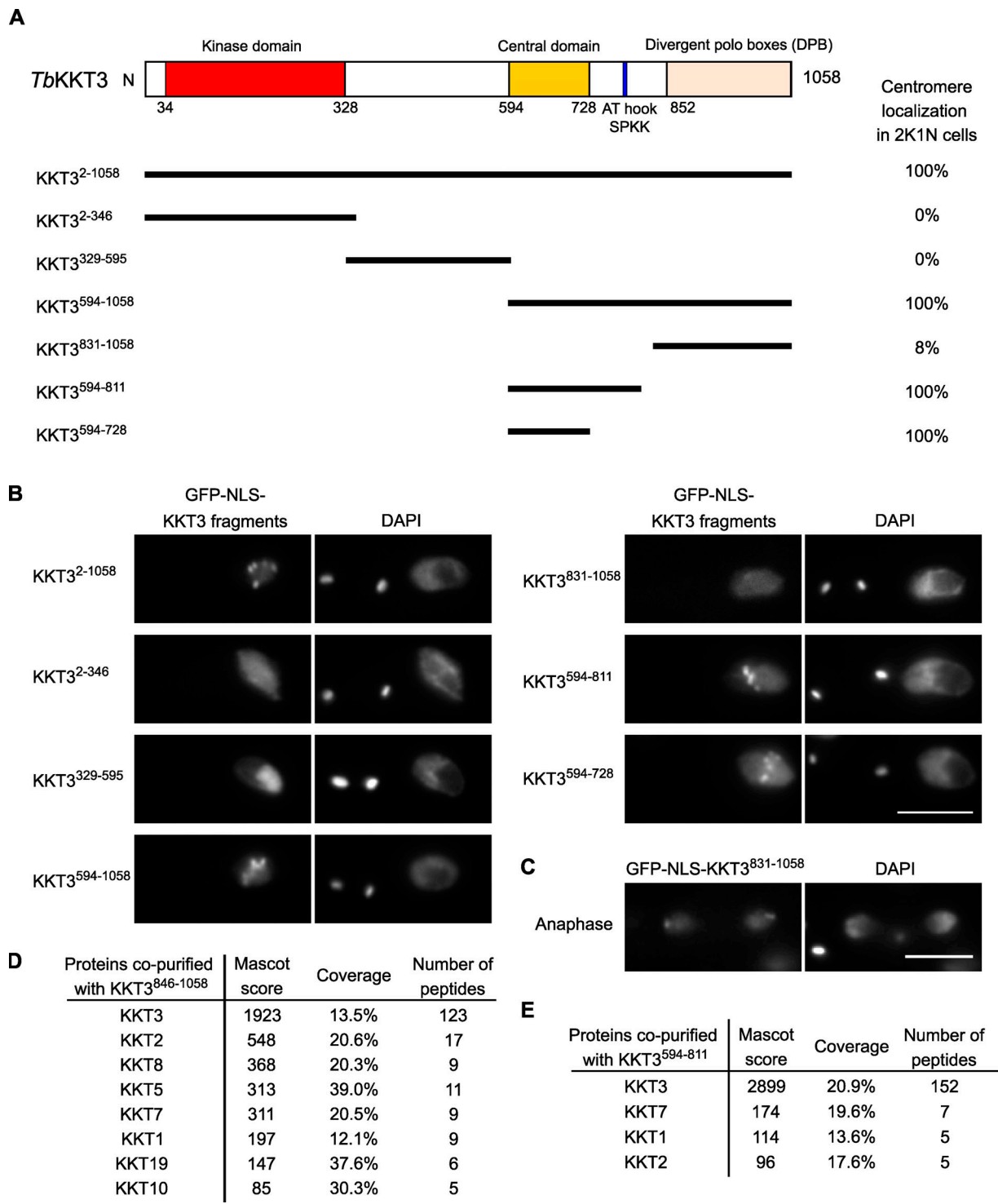

Figure 4. **KKT3 central domain is able to localize at centromeres constitutively in *T. brucei*. (A)** Schematic of the *T. brucei* KKT3 protein. Percentages of GFP-positive 2K1N cells (G2 to metaphase) that have kinetochore-like dots were quantified at 1 d after induction (*n* > 24 each). **(B)** Ectopically expressed *Tb*KKT3 fragments that contain the central domain form kinetochore-like dots. Inducible GFP-NLS fusion proteins were expressed with 10 ng/ml doxycycline. Cell lines: BAP291, BAP292, BAP379, BAP296, BAP378, BAP377, and BAP418. Scale bar, 5 µm. **(C)** *Tb*KKT3 DPB (831–1,058) forms kinetochore-like dots during anaphase (88% of 2K2N cells, *n* = 25). Note that kinetochore proteins typically localize near the leading edge of separating chromosomes during anaphase. Cell line: BAP296. Scale bar, 5 µm. **(D)** *Tb*KKT3 DPB copurifies with several kinetochore proteins. Cell line: BAP520. **(E)** *Tb*KKT3^594–811 does not copurify robustly with other kinetochore proteins. Cell line: BAP377. Inducible GFP-NLS fusion proteins were expressed with 10 ng/ml doxycycline, and immunoprecipitation was performed using anti-GFP antibodies. See Table S1 for all proteins identified by mass spectrometry.

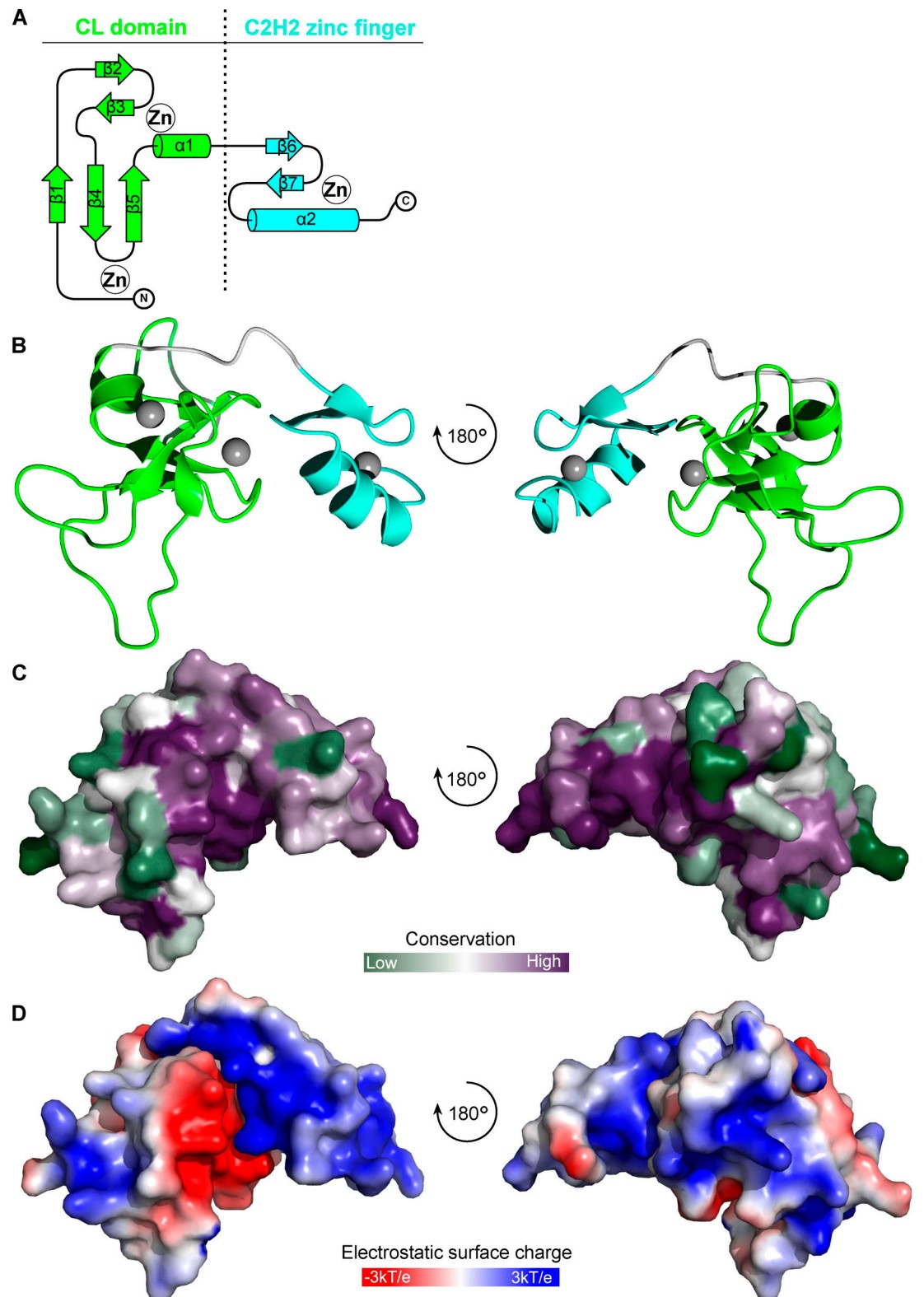

Figure 5. **Crystal structure of *B. saltans* KKT2 central domain reveals the presence of two zinc-binding domains. (A)** Topology diagram of the *Bs*KKT2 central domain showing the CL domain in green and C2H2-type zinc finger in cyan. **(B)** Cartoon representation of the *Bs*KKT2 central domain in two orientations. Zinc ions are shown in gray spheres. The structure is colored as in A. **(C)** Surface representation of the *Bs*KKT2 central domain colored according to sequence conservation using the ConSurf server (Landau et al., 2005; Ashkenazy et al., 2016). Structure orientation as in B. **(D)** Electrostatic surface potential of the *Bs*KKT2 central domain generated by Adaptive Poisson–Boltzmann Solver software (Jurrus et al., 2018). Structure orientation as in B.

Table 1. **Data collection and refinement statistics for *Bs*KKT2$^{572-668}$**

| Data collection | *Bs*KKT2 central domain |
| --- | --- |
| Beamline | Diamond I04 |
| Wavelength (Å) | 1.28297 |
| Space group (Z) | I222 (8) |
| Unit cell (cell edges in Å, cell angles in degrees) | 38.96, 53.51, 83.29, 90, 90, 90 |
| Resolution range (Å) | 45.02–1.8 (1.83–1.8) |
| Total no. of reflections | 37266 (925) |
| No. of unique reflections | 8158 (334) |
| Completeness (%) | 96.96 (77.86) |
| $R_{merge}$ | 0.10 (0.52) |
| $R_{pim}$ | 0.05 (0.33) |
| $CC_{1/2}$ | 0.99 (0.77) |
| $[I/\sigma(I)]$ | 8.65 (1.08) |
| Multiplicity | 4.57 (2.77) |
| Anomalous completeness (%) | 89.83 (37.37) |
| Anomalous multiplicity | 2.54 (1.93) |
| Overall *B* factor from Wilson plot (Å$^2$) | 22.4 |
| **Refinement** | |
| Resolution (Å) | 35.3–1.8 (2.0–1.8) |
| No. of reflections working set | 7,748 (390) |
| No. of reflections test set | 403 (18) |
| Final $R_{work}$ (%) | 19.5 (24.3) |
| Final $R_{free}$ (%) | 22.5 (23.4) |
| No. of protein atoms | 770 |
| No. of Zn atoms | 3 |
| No. of water atoms | 94 |
| No. of sulfate ions | 2 |
| Average B factor (Å$^2$) protein atoms | 30.45 |
| Average B factor (Å$^2$) Zn atoms | 25.31 |
| Average B factor (Å$^2$) water atoms | 44.43 |
| RMSD bond lengths (Å) | 0.008 |
| RMSD bond angles (°) | 0.96 |
| **Ramachandran plot** | |
| Most favored (%) | 97.89 |
| Allowed (%) | 2.11 |
| Disallowed | 0 |

Parentheses indicate the values relative to the highest-resolution shell.

Structural analysis of the C-terminal zinc-binding domain of *Bs*KKT2 revealed a classical C2H2-type zinc finger (Table S3). C2H2 zinc fingers are known to bind DNA, RNA, or protein (Krishna et al., 2003; Brayer and Segal, 2008). In most known cases, two or more C2H2 zinc fingers are used to recognize specific DNA sequences, which is typically achieved by specific interactions between the side chain of residues in positions –1, 2, 3, and 6 in the recognition α-helix—where –1 is the residue immediately preceding the α-helix—and DNA bases (Wolfe et al., 2000). Notably, some proteins with a single zinc finger can recognize specific DNA sequences (Omichinski et al., 1997; Dathan et al., 2002). The C-terminal zinc-binding domain of *Bs*KKT2 consists of one C2H2 domain (–1: Ser653; 2: Thr655; 3: Lys656; 6: Tyr659). The sequence alignment of the *Bs*KKT2 C2H2 zinc finger shows that residues at the positions –1 and 3 are highly conserved in trypanosomatids, while those at positions 2 and 6 are not (Fig. 8 A).

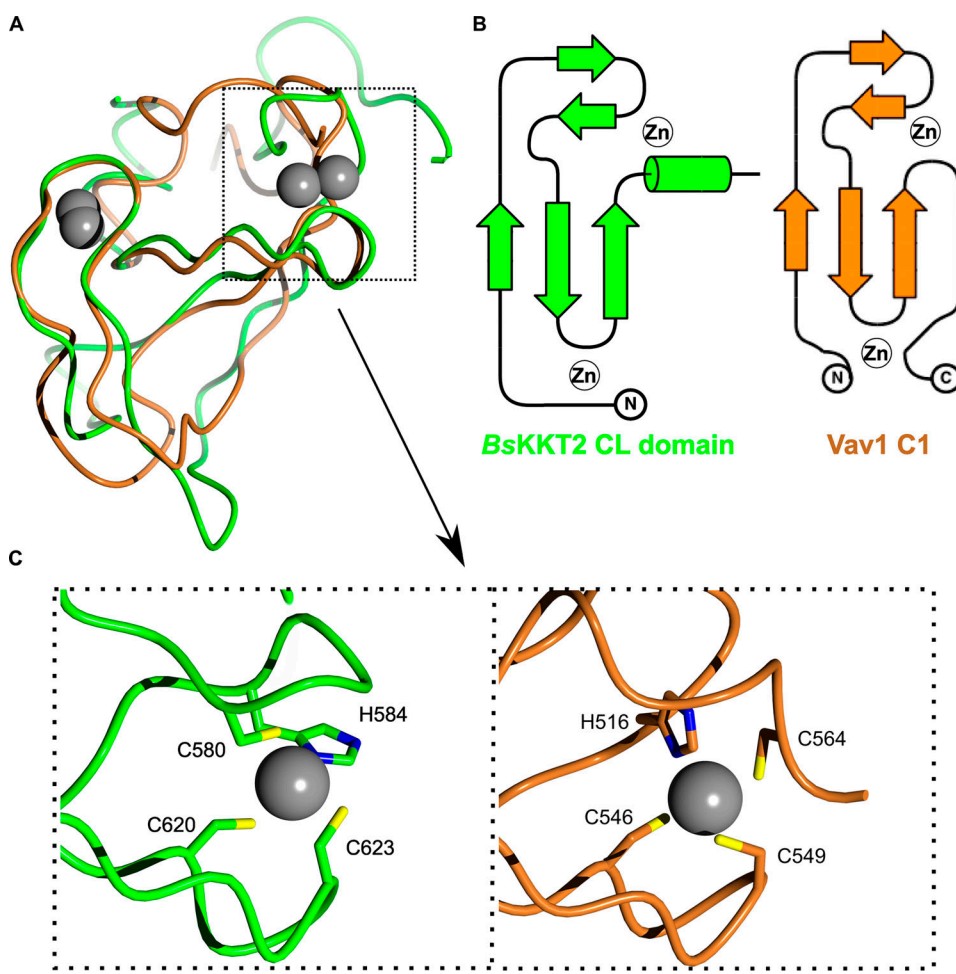

Figure 6. **B. saltans KKT2 CL is a unique domain. (A)** Structure superposition of *Bs*KKT2 CL domain in green and the Vav1 C1 domain in brown (Protein Data Bank accession no. 3KY9; Yu et al., 2010). Zinc ions are shown in gray spheres. **(B)** Topology diagram of *Bs*KKT2 CL domain and Vav1 C1 domain. **(C)** Close-up view showing a key difference in zinc coordination between *Bs*KKT2 CL domain and Vav1 C1 domain.

### The CL domain structure is conserved in *Perkinsela* KKT2a

We next asked whether the central domain structure is conserved among kinetoplastids. *Perkinsela* is a highly divergent endosymbiotic kinetoplastid that lives inside *Paramoeba* (Tanifuji et al., 2017). Our homology search identified three proteins that have similarity to KKT2 and KKT3. We call these *Perkinsela* proteins *Pk*KKT2a (XU18_4017), *Pk*KKT2b (XU18_0308), and *Pk*KKT2c (XU18_4564), because they tend to have higher sequence similarity to KKT2 than KKT3, especially in the kinase domain: 34.4% identical between *Pk*KKT2a and *Tb*KKT2 compared with 24.9% between *Pk*KKT2a and *Tb*KKT3; and 34.5% identical between *Pk*KKT2b and *Tb*KKT2 compared with 25.8% between *Pk*KKT2b and *Tb*KKT3. Interestingly, similarities among these *Perkinsela* proteins are higher than those between them and KKT2 or KKT3 in other kinetoplastids (e.g., the kinase domain is 59.6% identical between *Pk*KKT2a and *Pk*KKT2b; also see Fig. S3). *Pk*KKT2c does not have a kinase domain, like KKT20 in other kinetoplastids (Nerusheva and Akiyoshi, 2016). Our sequence alignment suggests that *Pk*KKT2a and *Pk*KKT2b have a CL-like domain but lack a C2H2 zinc finger (Fig. S3).

We determined the crystal structure of *Pk*KKT2a[551–679] at 2.9-Å resolution by Zn-SAD phasing (Fig. S2 and Table 2), which

confirmed the presence of a CL-like structure: Two β-sheets (residues 551–647), followed by an extended C-terminal α-helix (residues 648–679; Fig. 7, A and B). The CL-like domain of *Pk*KKT2a overlaps closely with that of *Bs*KKT2 (RMSD: 0.79 Å across 39 Cα), with the exception of some differences being localized to the loop insertion and the absence of a C2H2 domain in *Pk*KKT2a[551–679] (Fig. 7 C), consistent with our sequence analysis (Fig. S3). Taken together, our structures have revealed that the CL domain is conserved in *Bs*KKT2 and *Pk*KKT2a. Given the sequence similarity of KKT2 between *B. saltans* and other trypanosomatids (Fig. 8 A; 43.5% identical between *Bs*KKT2[564–680] and *Tb*KKT2[562–677]), it is likely that the unique CL domain structure is conserved among kinetoplastids.

### KKT2 has DNA-binding activity

Although *Perkinsela* KKT2a lacks a C2H2 zinc finger, our sequence analysis of the KKT2 central domain revealed a putative C2H2 zinc finger not only in trypanosomatids and bodonids, but also in one of Prokinetoplastina's KKT2-like proteins, PhM_4_m.86555 (Fig. S3; Tikhonenkov et al., 2021). Moreover, sequence analysis of KKT2/3 showed the presence of a DNA-binding SPKK motif (Suzuki, 1989) right after the C2H2 zinc

| Data collection | *Pk*KKT2a$^{551-679}$ | |
|---|---|---|
| **Beamline** | **Diamond I24** | **Diamond I03** |
| Wavelength (Å) | 0.96861 | 1.28272 |
| Space group (Z) | P6$_4$ (6) | P6$_4$ (6) |
| Unit cell (cell edges in Å, cell angles in degrees) | 113.84, 113.84, 46.01, 90, 90, 120 | 114.33, 114.33, 46.32 90, 90, 120 |
| Resolution range (Å) | 56.92–2.87 (2.92–2.87) | 99.01–3.80 (3.86–3.80) |
| Total no. of reflections | 153,076 (6844) | 68,026 (3882) |
| No. of unique reflections | 7,986 (357) | 3,535 (189) |
| Completeness (%) | 99.73 (91.77) | 100.00 (100.00) |
| $R_{merge}$ | 0.10 (1.04) | 0.31 (6.13) |
| $R_{pim}$ | 0.024 (0.240) | 0.073 (1.38) |
| CC$_{1/2}$ | 0.99 (0.45) | 0.99 (0.4) |
| [I/σ(I)] | 16.45 (2.14) | 12 (1.7) |
| Multiplicity | 19.17 (19.17) | 19.2 (20.5) |
| Anomalous completeness (%) | 99.6 (91.2) | 100 (100) |
| Anomalous multiplicity | 10 (9.9) | 10.2 (10.6) |
| Overall B factor from Wilson plot (Å$^2$) | 105.95 | 136.7 |
| **Refinement** | | |
| Resolution (Å) | 49–2.87 (3–2.87) | |
| No. of reflections working set | 7,408 (417) | |
| No. of reflections test set | 377 (16) | |
| Final $R_{work}$ (%) | 25.2 (36.6) | |
| Final $R_{free}$ (%) | 27.5 (51.2) | |
| No. of protein atoms | 832 | |
| No. of Zn atoms | 2 | |
| Average B factor (Å$^2$) protein atoms | 113.4 | |
| Average B factor (Å$^2$) Zn atoms | 100.6 | |
| RMSD bond lengths (Å) | 0.009 | |
| RMSD bond angles (°) | 1.11 | |
| **Ramachandran plot** | | |
| Most favored (%) | 90.9 | |
| Allowed (%) | 9.1 | |
| Disallowed | 0 | |

Parentheses indicate the values relative to the highest-resolution shell.

finger in many kinetoplastids, while an AT-hook motif is present within the CL-like domain of *Perkinsela* KKT2b (Fig. S3). These observations suggest that the central domain of KKT2 might have DNA-binding activity, perhaps stabilizing its localization at centromeres. To test this hypothesis and examine the importance of DNA-binding activity for the kinetochore localization of the KKT2 central domain, we performed fluorescence polarization assays using fluorescently labeled DNA probes. Unfortunately, we were unable to obtain reliable data for *T. brucei* and *B. saltans* KKT2 central domains due to fluorophore quenching. We therefore focused on *Pk*KKT2a$^{551-679}$ that has a CL-like domain. Our fluorescence polarization assay showed that *Pk*KKT2a$^{551-679}$

has DNA-binding activity with a dissociation constant ($K_d$) of ~500 nM on three DNA probes that have different GC contents (50 mer CEN is 50-bp DNA sequence from the CIR147 centromere repeat in *T. brucei*; Fig. S4 A). To assess the importance of the CL domain for DNA binding, we next performed a fluorescence polarization assay for *Pk*KKT2a$^{551-679}$, which has mutations in zinc-coordinating residues (C646 and C649, corresponding to C616 and C619 in *Tb*KKT2) and found that it has a similar DNA-binding affinity compared with WT *Pk*KKT2a$^{551-679}$ (Fig. S4 B). As a comparison, we used a well-characterized zinc finger (designed zinc finger; Fig. S2 C) that binds a specific DNA sequence (designed DNA; Jantz and Berg, 2010).

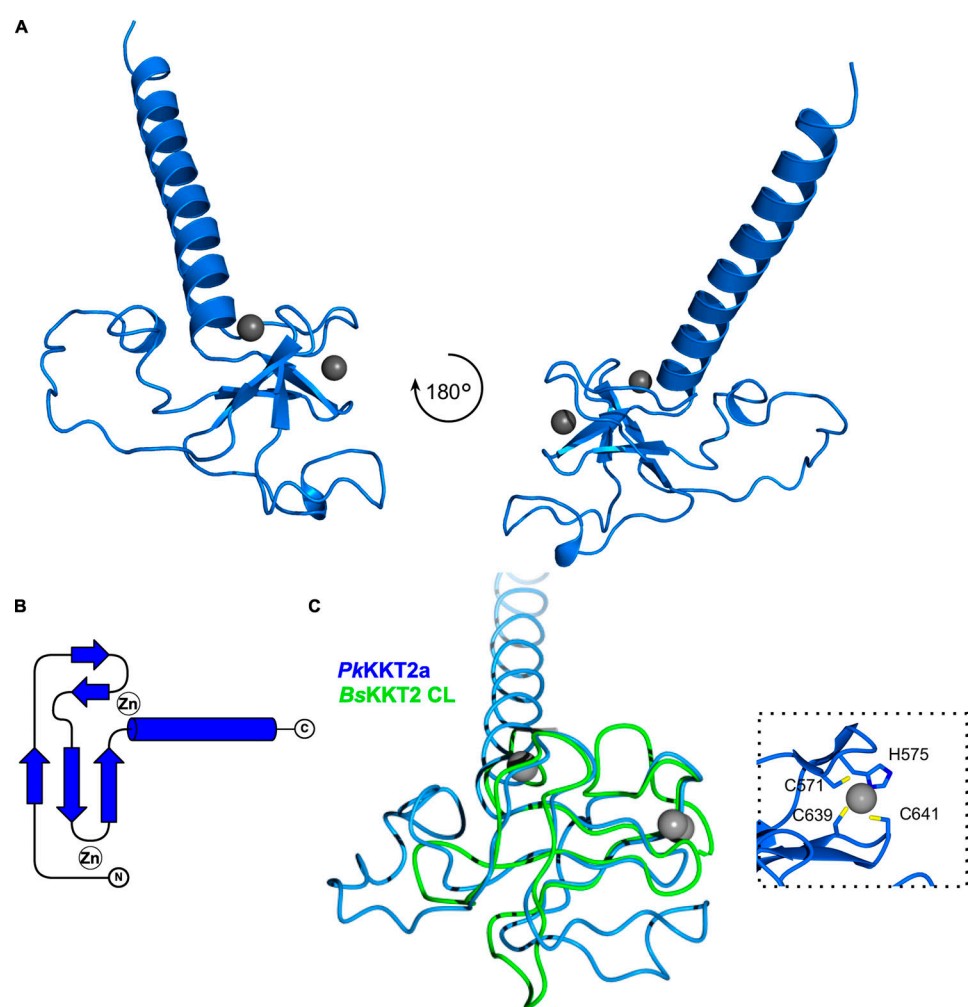

**Figure 7. Crystal structure of *Perkinsela* KKT2a$^{551-679}$ highlights conservation of CL domain. (A)** Cartoon representation of *Pk*KKT2a$^{551-679}$ in two orientations. Zinc ions are shown in gray spheres. **(B)** Topology diagram of *Pk*KKT2a$^{551-679}$ structure. **(C)** Structure superposition of *Pk*KKT2a$^{551-679}$ and *Bs*KKT2 CL domain, showing that the core of the structure is conserved. Variations between the two structures are due to sequence insertions within CL and the absence of of the C2H2 zinc finger at the C terminus in *Pk*KKT2a$^{551-679}$. Inset shows a close-up view of zinc ion coordination mediated by the N-terminal residues C571 and H575 and the C-terminal residues C639 and C641.

This protein bound its optimal DNA sequence with a $K_d$ of 8 nM, while it had weaker affinity for 20-bp and 50-bp probes from the CIR147 centromere sequence (Fig. S4 C). These results show that *Pk*KKT2a$^{551-679}$ has CL domain–independent DNA-binding activity. It is possible that *Pk*KKT2a$^{551-679}$ has higher DNA-binding activity for yet-to-be-identified *Perkinsela* centromere DNA sequences.

In another attempt to study DNA-binding activity of *Tb*KKT2$^{562-677}$ (Fig. S2 D), we performed electrophoretic mobility shift assay (EMSA) against unlabeled DNA probes. Our assay showed that *Tb*KKT2$^{562-677}$ has weak DNA-binding activity for all the probes tested (Fig. S4 D). To assess the importance of the CL domain for DNA binding, we next performed an EMSA for *Tb*KKT2$^{562-677}$ that has mutations in zinc-coordinating residues (C597 and C600) and found that it has similar DNA-binding activity as WT *Tb*KKT2$^{562-677}$ (Fig. S4 E). These results show that, similar to *Pk*KKT2a$^{551-679}$, *Tb*KKT2$^{562-677}$ weakly binds DNA in a sequence-independent manner and that the CL domain is largely dispensable for its DNA-binding activity.

**The CL domain of KKT2 is important for long-term viability in *T. brucei***

To examine the functional relevance of the CL domain and C2H2 zinc finger, we tested their mutants in *T. brucei*. We first made various mutants in full-length *Tb*KKT2 and found that all mutants localized at kinetochores (Fig. S5 A). Because *Tb*KKT2 has multiple domains that can independently localize at kinetochores (Fig. 3), we next expressed mutants in our ectopic expression of the central domain (*Tb*KKT2$^{562-677}$). We found that mutations in zinc-coordinating residues of the CL domain (C576A, H580A, C597A, C600A, C616A, and C619A) all abolished kinetochore localization (Fig. 8, A and B). In contrast, similar mutations in the C2H2-type zinc finger (C640A and C643A) did not affect the localization.

To gain insight into how the KKT2 CL domain may promote kinetochore localization, we analyzed conservation and electrostatic potential of the *Bs*KKT2 CL domain surface residues to identify possible patches that may be involved in this process. Our analysis revealed a highly conserved acidic patch centered

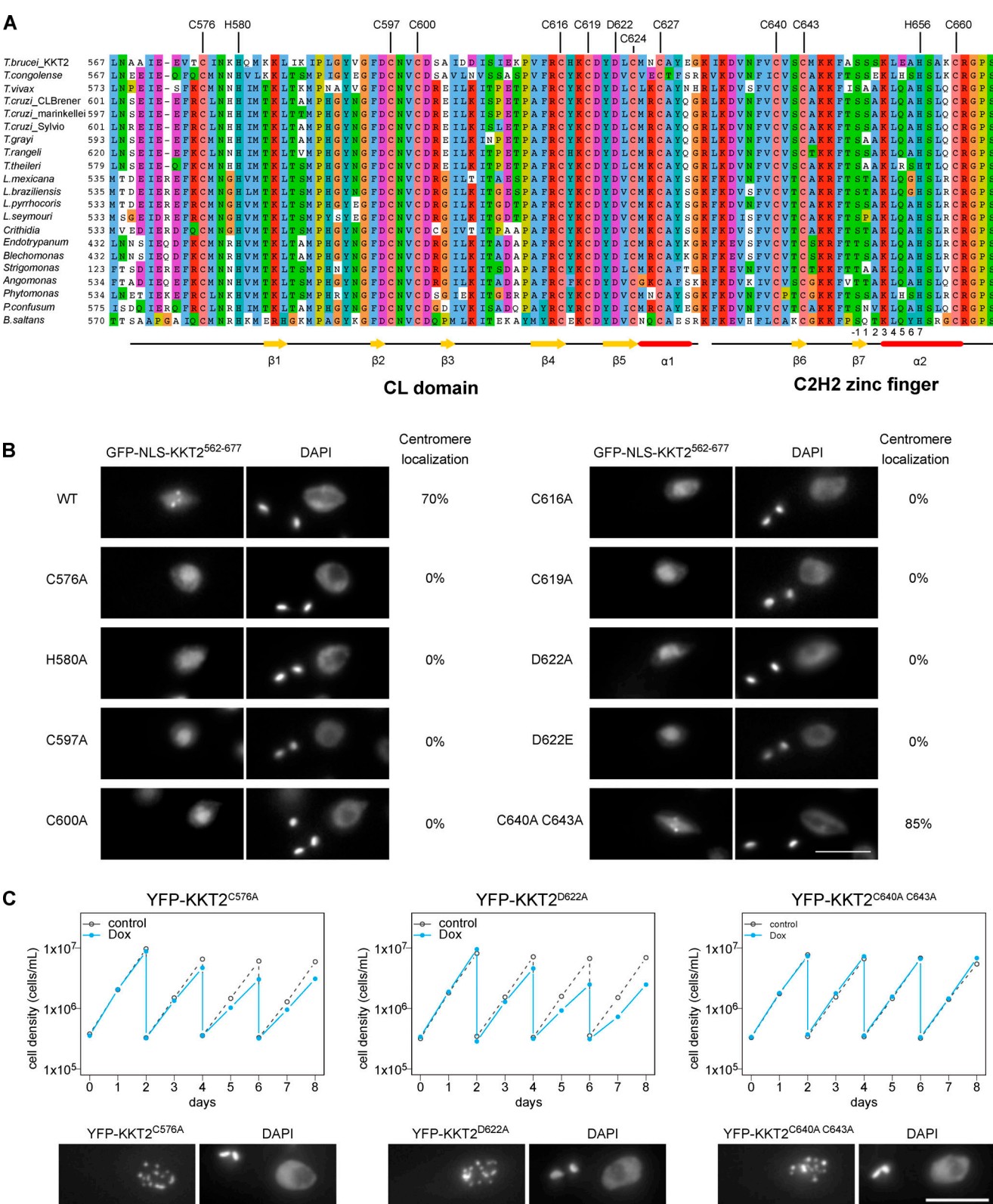

Figure 8. **KKT2 CL domain is critical for the centromere localization in *T. brucei*. (A)** Multiple sequence alignment of KKT2. Residues that coordinate zinc ions, the conserved aspartic acid residue, and secondary structures of *Bs*KKT2 are shown. Numbers below the alignment are for the C2H2 zinc finger's α-helix, where −1 is the residue immediately preceding the helix. **(B)** Kinetochore localization of *Tb*KKT2$^{562-677}$ depends on the CL domain, but not on C2H2 zinc finger. Percentages of GFP-positive 2K1N cells that have kinetochore-like dots were quantified at 1 d after induction (*n* = 40 each). Inducible GFP-NLS fusion proteins were expressed with 10 ng/ml doxycycline. Cell lines: BAP457, BAP1700, BAP1702, BAP1710, BAP1712, BAP1715, BAP1717, BAP1649, BAP1719, and BAP1837. **(C)** *Tb*KKT2 C576A and D622A mutants localize at kinetochores but fail to support normal cell growth. One allele of *Tb*KKT2 was mutated and tagged with an N-terminal YFP tag, and the other allele was depleted using RNAi-mediated knockdown by targeting the 5′UTR of the *Tb*KKT2 transcript. Top: Cells were

diluted every 2 d, and cell growth was monitored for 8 d upon induction of RNAi. Controls are uninduced cell cultures. Similar results were obtained for at least three clones of *Tb*KKT2 mutants. Bottom: Example of cells expressing the *Tb*KKT2 mutants before RNAi induction, showing that they localize at kinetochores ($n$ > 150 each; also see Fig. S5, C and D). Maximum intensity projections are shown. RNAi was induced with 1 μg/ml doxycycline (Dox). Cell lines: BAP1789, BAP1779, and BAP1786. Scale bars, 5 μm.

around residue *Bs*KKT2 D626 (Fig. 5, C and D; and Fig. S3). Interestingly, this aspartic acid is strictly conserved in all KKT2 and KKT3 proteins (Fig. S3). To test the importance of this residue, we mutated the corresponding residue in *T. brucei* and found that *Tb*KKT2$^{562–677}$ with either D622A or D622E failed to localize at kinetochores (Fig. 8 B). It is noteworthy that even such a subtle mutation (from D to E, which is unlikely to change the electrostatic surface charge of the patch) abolished the localization of *Tb*KKT2$^{562–677}$. Taken together, our results show that the CL domain, but not the C2H2 zinc finger, is important for the kinetochore localization of the *Tb*KKT2 central domain.

To test the importance of the *Tb*KKT2 central domain for cell viability, we next performed rescue experiments. We replaced one allele of *Tb*KKT2 with an N-terminally YFP-tagged *Tb*KKT2 construct that has either WT or mutant versions of the central domain and performed RNAi against the 5′UTR of the *Tb*KKT2 transcript to knockdown the untagged allele of *Tb*KKT2 (Fig. S5 B; Ishii and Akiyoshi, 2020). As expected, mutants in the CL domain (C576A and D622A) and the C2H2 zinc finger (C640A and C643A) both localized at kinetochores (Fig. 8 C; and Fig. S5, C and D). Upon induction of RNAi, however, the CL domain mutants failed to support normal cell growth after day 4, while the C2H2 zinc finger mutants rescued the growth defects (Fig. 8 C). These data confirm the importance of the CL domain for the function of *Tb*KKT2 in vivo.

**Localization of KKT3 depends on the central domain in *T. brucei***
The central domain of *Tb*KKT3 can localize at kinetochores throughout the cell cycle (Fig. 4). The sequence similarity of the central domain between KKT2 and KKT3 (29.9% identical between *Tb*KKT2$^{575–660}$ and *Tb*KKT3$^{645–727}$) suggested that *Tb*KKT3 likely consists of two domains that correspond to the CL domain and the C2H2 zinc finger present in KKT2 (Fig. S3). Consistent with this prediction, mutating *Tb*KKT3 residues that align with zinc-coordinating histidine or cysteine residues in the CL domain of KKT2 abolished the kinetochore localization of the ectopically expressed full-length *Tb*KKT3 protein (Fig. 9, A and B). We also found that the conserved aspartic acid D692 (Fig. S3) was essential for kinetochore localization, because D692A and D692E mutants both abolished kinetochore localization of KKT3 (Fig. 9 B). In contrast, mutations in the *Tb*KKT3 C2H2 zinc finger (C707A and C710A) did not affect kinetochore localization.

We next performed rescue experiments by replacing one allele of *Tb*KKT3 with a C-terminally YFP-tagged construct that has either WT or mutant versions of the central domain and performed RNAi against the 3′UTR of *Tb*KKT3 to knock down the untagged allele of *Tb*KKT3 (Fig. S5 B). We first confirmed that *Tb*KKT3 CL domain mutants (C668A, C671A, and D692A) were unable to localize at kinetochores, while the *Tb*KKT3 C2H2 zinc finger mutant (C707A and C710A) localized normally (Fig. 9 C and Fig. S5 E). Upon induction of RNAi, CL mutants failed to

rescue the growth defect, showing that kinetochore localization is essential for the *Tb*KKT3 function (Fig. 9 C). In contrast, the *Tb*KKT3 C2H2 zinc finger mutant supported normal cell growth. These data show that the *Tb*KKT3 CL domain is essential for the localization and function of *Tb*KKT3.

## Discussion

A major open question concerning the biology of kinetoplastids is how these organisms assemble kinetochores specifically at centromeres using a unique set of kinetochore proteins. Studies in other eukaryotes have shown that constitutively localized kinetochore proteins, such as CENP-A and CENP-C, play crucial roles in kinetochore specification and assembly (French and Straight, 2017; Hamilton and Davis, 2020; Kixmoeller et al., 2020). Among the six proteins that constitutively localize at kinetochores in *T. brucei* (KKT2, KKT3, KKT4, KKT20, KKT22, and KKT23), we previously showed that KKT4 is important for the kinetochore localization of KKT20, but not many other proteins, including KKT2 and KKT3 (Llauró et al., 2018). In this study, we show that KKT2 and KKT3 are important for recruiting multiple kinetochore proteins, including KKT1, KKT4, KKT14, KKT22, and KKT23, while localization of KKT2 and KKT3 is independent from various kinetochore proteins (Fig. 10). Together with the fact that KKT2 and KKT3 have DNA-binding motifs, these results support the hypothesis that KKT2 and KKT3 locate at the base of kinetoplastid kinetochores and play crucial roles in recruiting other kinetochore proteins. KKT2/3 share common ancestry with polo-like kinases (Nerusheva and Akiyoshi, 2016). In addition to an N-terminal protein kinase domain and C-terminal divergent polo boxes, they have a central domain that is highly conserved among kinetoplastids. Polo boxes are the protein–protein interaction domains found in polo-like kinases that often must be phosphorylated to enable protein–protein interactions (Elia et al., 2003; Zitouni et al., 2014); therefore, these domains are prime candidates as nucleation sites on which the kinetoplastid kinetochore could be assembled. Interestingly, those residues in human PLK1 that play key roles in phospho-peptide recognition are not present in the divergent polo boxes of KKT2 and KKT3 (Nerusheva and Akiyoshi, 2016). It will be important to identify which proteins directly interact with KKT2/3 to understand the mechanism of kinetochore assembly. It will also be important to examine whether kinase activities of KKT2/3 are important for kinetochore assembly or any kinetochore function.

By ectopically expressing fragments of KKT2 and KKT3 in *T. brucei*, we established that their central domains can localize specifically at centromeres. The crystal structure of the *B. saltans* KKT2 central domain revealed a unique structure, which consists of two distinct zinc-binding domains, the CL domain and C2H2-type zinc finger. It is likely that the central domain of

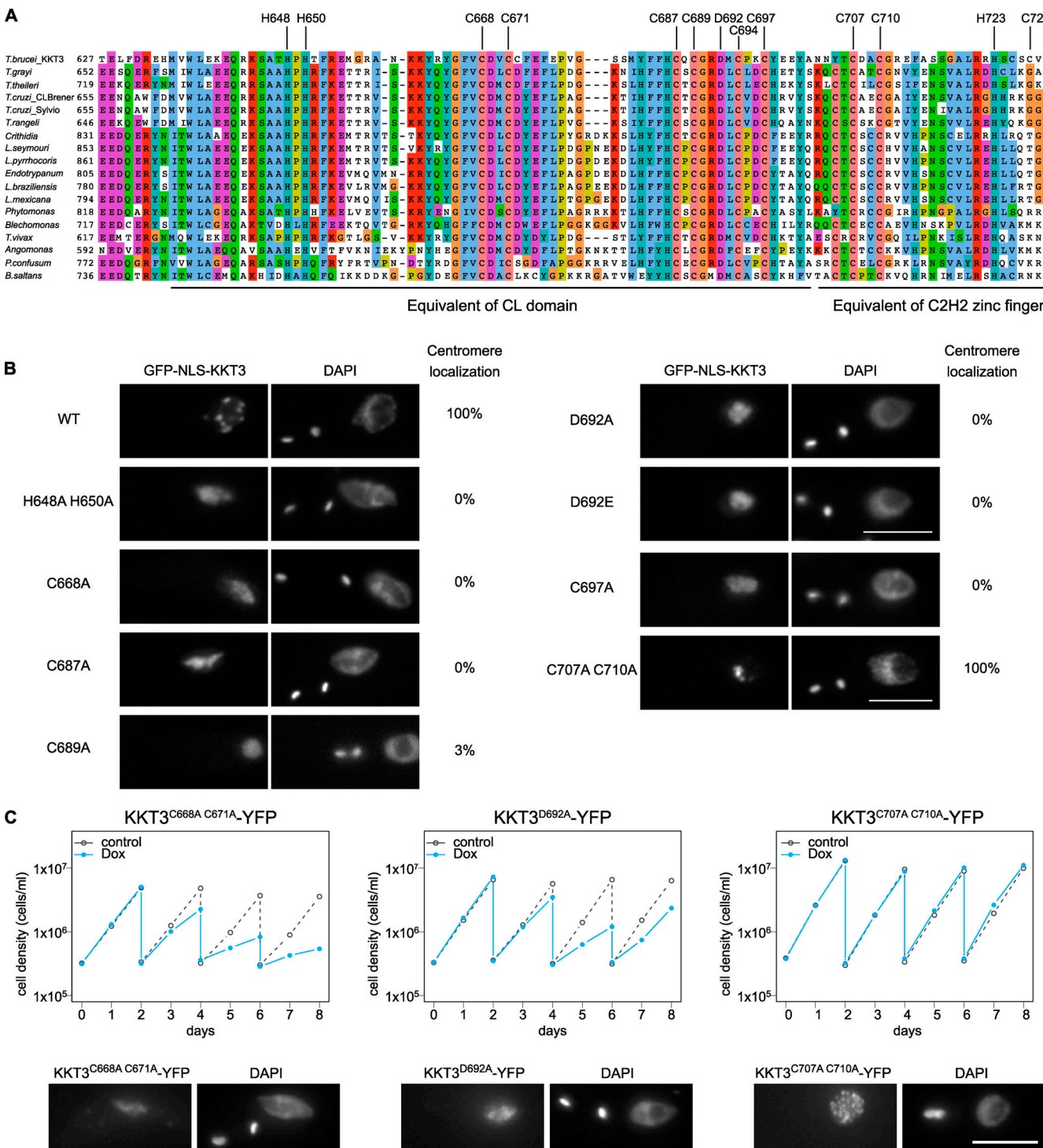

Figure 9. **Kinetochore localization of KKT3 depends on the central domain in *T. brucei*. (A)** Multiple sequence alignment of KKT3. Residues that are expected to coordinate zinc ions as well as the conserved aspartic acid residue are shown. **(B)** Percentage of GFP-positive cells that have kinetochore-like dots were quantified at 1 d after induction (*n* > 22 each). Inducible GFP-NLS fusion proteins were expressed with 10 ng/ml doxycycline. Cell lines: BAP291, BAP359, BAP360, BAP446, BAP447, BAP1721, BAP1722, BAP362, and BAP341. **(C)** *Tb*KKT3 C668A/C671A and D692A mutants do not localize at kinetochores and fail to support normal cell growth, while *Tb*KKT3 C707A/C710A mutant is functional. One allele of *Tb*KKT3 was mutated and tagged with a C-terminal YFP tag, and the other allele was depleted using RNAi-mediated knockdown by targeting the 3′UTR of the *Tb*KKT3 transcript. Top: Cells were diluted every 2 d, and cell growth was monitored for 8 d upon induction of RNAi. Similar results were obtained for at least three clones of *Tb*KKT3 mutants. Controls are uninduced cell cultures. Bottom: Example of cells expressing the *Tb*KKT3 mutants before RNAi induction, showing that *Tb*KKT3^C668A C671A and *Tb*KKT3^D692A do not localize at kinetochores while *Tb*KKT3^C707A C710A localizes normally (*n* > 90 each; also see Fig. S5 E). Maximum intensity projections are shown. RNAi was induced with 1 µg/ml doxycycline (Dox). Cell lines: BAP1791, BAP1793, and BAP1783. Scale bars, 5 µm.

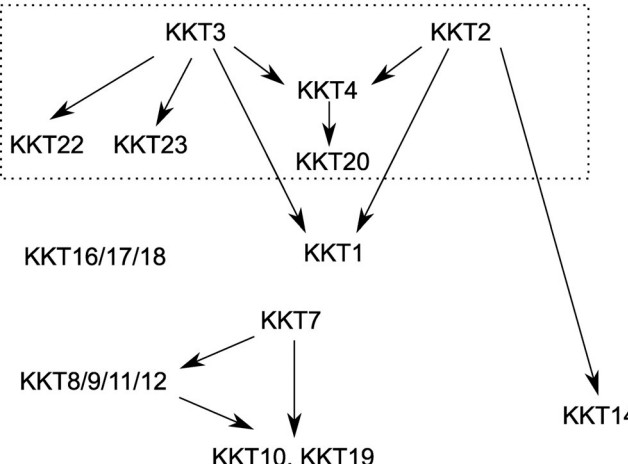

**Constitutive kinetochore proteins**

Figure 10.    **Summary of the kinetochore assembly pathway in *T. brucei*.** Arrows indicate localization dependency identified in this study and our previous study (Ishii and Akiyoshi, 2020). Proteins in the dotted box are constitutive kinetochore proteins, while others are transient kinetochore proteins. KKT2 (this study) and KKT4 (Ludzia et al., 2021) have been shown to have DNA-binding activity, and KKT3 has putative DNA-binding motifs. KKT4 is the only known microtubule-binding kinetochore protein in *T. brucei* (Llauró et al., 2018). Direct protein–protein interactions among kinetoplastid kinetochore proteins remain unknown except for KKT16/17/18 that form the KKT16 complex (Tromer et al., 2021), KKT8/9/11/12 that form the KKT8 complex, and KKT7 that binds KKT10 and KKT19 (Ishii and Akiyoshi, 2020). Known substrates of the KKT10/19 kinases include KKT4 and KKT7 (Ishii and Akiyoshi, 2020), while substrates of KKT2/3 remain unknown.

*T. brucei* KKT2 has a similar structure based on the high sequence similarity between the *Bs*KKT2 and *Tb*KKT2 proteins (43.5% identical between *Bs*KKT2$^{564–680}$ and *Tb*KKT2$^{562–677}$). Importantly, mutational analyses of *Tb*KKT2 revealed that the CL domain is important for the localization of the central domain, while the C2H2 zinc finger is not. Furthermore, although full-length *Tb*KKT2 CL mutants localized at kinetochores (likely due to interactions with other kinetochore proteins via other domains of *Tb*KKT2), they were not fully functional. Taken together, these data have established that the CL domain is essential for the function of *Tb*KKT2, which is consistent with the presence of CL, but not C2H2 zinc finger, in *Perkinsela* KKT2a. Given that the CL domain mutants of *Tb*KKT2 and *Pk*KKT2a still have DNA-binding activity, it is unlikely that centromere localization of the KKT2 central domain relies solely on its DNA-binding activity.

It remains unclear whether the structure of the central domain is conserved between KKT2 and KKT3. Nonetheless, our functional studies showed that the equivalent domain of CL in *Tb*KKT3 was also essential for the kinetochore localization and function, while the equivalent domain of C2H2 zinc finger was not, showing that the functional importance of CL is conserved in KKT3. It will be important to obtain KKT3 central domain structures to reveal structural similarity or difference between KKT2 and KKT3. It is noteworthy that all identified KKT2/3 homologues in deep-branching Prokinetoplastina have greater similarity to KKT2 than KKT3. We speculate that ancestral

kinetoplastids had only a KKT2-like protein(s) that performed all necessary functions, and that KKT3 in trypanosomatids and bodonids represents a product of gene duplication that became specialized in certain functions, such as more efficient centromere localization by its central domain compared with KKT2.

It remains unclear how kinetoplastids specify kinetochore positions. The fact that KKT2/3 central domains manage to localize at centromeres suggests that they are able to recognize something special at centromeres. What might be a unique feature at centromeres in kinetoplastids that lack CENP-A? Histone variants are one possibility. *T. brucei* has four histone variants: H2AZ, H2BV, H3V, and H4V; however, none of them is specifically enriched at centromeres (Lowell and Cross, 2004; Lowell et al., 2005; Siegel et al., 2009), and histone chaperones did not copurify with any kinetochore protein (Akiyoshi and Gull, 2014). Alternatively, there might exist certain posttranslational modifications on histones or DNA specifically at centromeres (e.g., phosphorylation, methylation, acetylation, ubiquitination, or sumoylation). The KKT2 CL domain has a highly conserved acidic patch that might act as a reader for such modifications. Although there is no known histone or DNA modification that occurs specifically at centromeres, KKT2/3 have a protein kinase domain and KKT23 has a Gcn5-related N-acetyltransferase domain (Nerusheva et al., 2019). It will be important to examine whether these enzymatic domains are important for proper recruitment of KKT2/3 central domains. Another unique feature at centromeres is the presence of kinetochore proteins, which could potentially recruit newly synthesized kinetochore components by direct protein–protein interactions. Finally, it is important to note that it remains unclear whether kinetoplastid kinetochores build upon nucleosomes. It is formally possible that the KKT2/3 central domains directly bind DNA and form a unique environment at centromeres. Understanding how the KKT2/3 central domains localize specifically at centromeres will be key to elucidating the mechanism of how kinetoplastids specify kinetochore positions in the absence of CENP-A.

## Materials and methods
### Trypanosome cells and plasmids
All trypanosome cell lines, plasmids, primers, and synthetic DNA used in this study are listed in Table S4. All trypanosome cell lines used in this study were derived from *T. brucei* SmOxP927 procyclic form cells (TREU 927/4 expressing T7 RNA polymerase and the tetracycline repressor to allow inducible expression; Poon et al., 2012). Cells were grown at 28°C in SDM-79 medium supplemented with 10% (vol/vol) heat-inactivated FCS (Brun and Schönenberger, 1979). Endogenous YFP tagging was performed using the pEnT5-Y vector (Kelly et al., 2007). Endogenous tdTomato tagging was performed using pBA148 (Akiyoshi and Gull, 2014) and its derivatives. Inducible expression of GFP-NLS fusion proteins was performed using pBA310 (Nerusheva and Akiyoshi, 2016).

To make pBA1711 (KKT3 3′UTR hairpin RNAi construct targeting the KKT3 transcript from stop codon to +370 bp), BAG95 synthetic DNA fragment was digested with HindIII/BamHI and

subcloned into the HindIII/BamHI sites of pBA310. pBA2052 (KKT2/3 double-hairpin RNAi construct targeting KKT2 5′UTR from −342 bp to start codon as well as KKT3 3′UTR from stop codon to +370 bp) was made with BAG129 as above. Similarly, pBA861 (KKT1 hairpin RNAi targeting 1,351–1,776 bp) was made with BAG24, pBA864 (KKT6 hairpin RNAi targeting 12–499 bp) with BAG27, pBA869 (KKT14 hairpin RNAi targeting 859–1,274 bp) with BAG32, pBA1316 (KKT8 3′UTR hairpin RNAi targeting from +31 to +446 bp) with BAG79, pBA1845 (KKT22 hairpin RNAi targeting 466 bp to stop codon) with BAG105, pBA1997 (KKT24 hairpin RNAi targeting 1,001–1,400 bp) with BAG115, and pBA2021 (KKT23 hairpin RNAi targeting 635–1,047 bp) with BAG127. To make pBA1091 (KKIP1 RNAi), 297–856 bp of KKIP1 coding sequence was amplified with primers BA1541/BA1543 and cloned into p2T7-177 using BamHI/HindIII sites (Wickstead et al., 2002). To make pBA1807 (C-terminal YFP tagging of KKT3), 4–3,174 bp of KKT3 coding sequence and 250 bp of 3′UTR were amplified with primers BA2351/BA2352 and BA2353/BA2354, digested with HindIII/NotI and NotI/SpeI, respectively, and cloned into the pEnT5-Y using HindIII/SpeI sites. Details of other plasmids are described in Table S4. Site-directed mutagenesis was performed using primers and template plasmids listed in Table S4. All constructs were sequence verified.

Plasmids linearized by NotI were transfected into trypanosomes by electroporation into an endogenous locus (pEnT5-Y derivatives and pBA148/pBA192/pBA892 derivatives) or 177-bp repeats on minichromosomes (pBA310 derivatives and p2T7-177 derivatives). Concentrations of drugs used were as follows: 5 µg/ml phleomycin, 25 µg/ml hygromycin, 10 µg/ml blasticidin, 30 µg/ml G418, and 1 µg/ml puromycin. To obtain endogenously tagged clonal strains, transfected cells were selected by the addition of appropriate drugs and cloned by dispensing dilutions into 96-well plates. Clones that express mutant versions of KKT2 or KKT3 from the endogenous locus were screened by Sanger sequencing of genomic DNA. Expression of GFP-NLS fusion proteins (pBA310 derivatives) was induced by the addition of doxycycline (10 ng/ml). RNAi was induced by the addition of doxycycline (1 µg/ml).

**Microscopy**
To analyze fluorescently tagged proteins, cells were washed once with PBS, settled onto glass slides, and fixed with 4% paraformaldehyde in PBS for 5 min (Nerusheva and Akiyoshi, 2016; Ishii and Akiyoshi, 2020). Cells were then permeabilized with 0.1% NP-40 in PBS for 5 min and embedded in mounting media (1% wt/vol 1,4-diazabicyclo[2.2.2]octane, 90% glycerol, 50 mM sodium phosphate, pH 8.0) containing 100 ng/ml DAPI. Images were captured on a DeltaVision fluorescence microscope (Applied Precision) with softWoRx (version 5.5) housed in Micron Oxford. Fluorescent images were captured at RT with a CoolSNAP HQ camera using a 60× objective lens (1.42 NA) or 100× objective lens (1.4 NA) and processed in Fiji software (Schneider et al., 2012). Typically, 25 optical slices spaced 0.2-µm apart were collected. Maximum intensity projection images were generated by Fiji. Kinetochore localization of endogenously tagged kinetochore proteins or ectopically expressed KKT2/3 fragments were examined manually by quantifying the number

of cells that clearly had detectable kinetochore-like dots at indicated cell cycle stages.

**Immunoprecipitation and mass spectrometry**
To identify interaction partners of KKT2 or KKT3 fragments fused with GFP-NLS, we performed immunoprecipitation using anti-GFP antibodies and identified copurifying proteins by mass spectrometry (Akiyoshi and Gull, 2014; Ishii and Akiyoshi, 2020). Typically, 400-ml cultures of asynchronously growing cells were grown to ∼2 × 10$^6$ cells/ml and expression of GFP-NLS fusion proteins was induced with 10 ng/ml doxycycline for 24 h. Cells were pelleted by centrifugation (900 g, 10 min), washed once with PBS, and extracted in PEME (100 mM Pipes-NaOH, pH 6.9, 2 mM EGTA, 1 mM MgSO$_4$, and 0.1 mM EDTA) with 1% NP-40 and protease inhibitors (10 µg/ml leupeptin, 10 µg/ml pepstatin, 10 µg/ml E-64, and 0.2 mM PMSF) and phosphatase inhibitors (1 mM sodium pyrophosphate, 2 mM Na-β-glycerophosphate, 0.1 mM Na$_3$VO$_4$, 5 mM NaF, and 100 nM microcystin-LR) for 5 min at RT, followed by centrifugation (1,800 g, 15 min). Samples were kept on ice from this point on. The pelleted fractions that contain kinetochore proteins were resuspended in modified buffer H (BH0.15: 25 mM Hepes, pH 8.0, 2 mM MgCl$_2$, 0.1 mM EDTA, pH 8.0, 0.5 mM EGTA, pH 8.0, 1% NP-40, 150 mM KCl, and 15% glycerol) containing protease inhibitors and phosphatase inhibitors. Samples were sonicated to solubilize kinetochore proteins (12 s, three times with 1-min interval on ice). 12 µg of mouse monoclonal anti-GFP antibodies (11814460001; Roche) that had been preconjugated with 60 µl slurry of Protein-G magnetic beads (10004D; Dynal) with dimethyl pimelimidate (Unnikrishnan et al., 2012) were incubated with the extracts for 2.5 h with constant rotation, followed by four washes with modified BH0.15 containing protease inhibitors, phosphatase inhibitors, and 2 mM DTT. Beads were further washed three times with preelution buffer (50 mM Tris-HCl, pH 8.3, 75 mM KCl, and 1 mM EGTA). Associated proteins were gently eluted from the beads by agitation in 60 µl of elution buffer (0.1% RapiGest [186001860; Waters] and 50 mM Tris-HCl, pH 8.3) for 25 min at RT. Samples were incubated at 100°C for 5 min. Proteins were reduced with 5 mM DTT at 37°C for 30 min and alkylated with 10 mM iodoacetamide at 37°C for 30 min. The reaction was quenched by adding 10 mM DTT at 37°C for 30 min, and 100 µl of 20 mM Tris-HCl (pH 8.3) was added. Proteins were digested overnight at 37°C with 0.2 µg trypsin (Promega). Formic acid was then added to 2% and the samples were incubated at 37°C for 30 min to cleave RapiGest, followed by centrifugation for 10 min. The supernatant was desalted over a C18 column and analyzed by electrospray tandem mass spectrometry over a 60-min gradient using Q-Exactive (Thermo Fisher Scientific) at the Advanced Proteomics Facility (University of Oxford). Peptides were identified by searching tandem mass spectrometry spectra against the *T. brucei* protein database with Mascot (version 2.5.1; Matrix Science) with carbamidomethyl cysteine as a fixed modification. Up to two missed cleavages were allowed. Oxidation (Met), phosphorylation (Ser, Thr, and Tyr), and acetylation (Lys) were searched as variable modifications. Mass tolerances for mass spectrometry and tandem mass spectrometry peak identifications were 20 ppm and 0.02 D, respectively. Proteins identified with at

least two peptides were considered significant and listed in Table S1.

## Multiple sequence alignment

Protein sequences and accession nos. for KKT2 and KKT3 homologues were retrieved from TriTryp database (Aslett et al., 2010), Wellcome Sanger Institute (https://www.sanger.ac.uk/), UniProt (UniProt Consortium, 2019), or a published study (Butenko et al., 2020). Searches for KKT2/3 homologues in Prokinetoplastina and Bodonida were done using hmmsearch on its predicted proteome using manually prepared KKT2/3 hmm profiles (HMMER version 3.0; Eddy, 1998). Multiple sequence alignment was performed with MAFFT (L-INS-i method, version 7; Katoh et al., 2019) and visualized with the Clustalx coloring scheme in Jalview (version 2.10; Waterhouse et al., 2009).

## Protein expression and purification

Multiple sequence alignment, together with secondary structure predictions of the KKT2 central domain, were used to design constructs in *B. saltans* and *Perkinsela*. To make pBA1660 ($BsKKT2^{572-668}$ with an N-terminal tobacco etch virus–cleavable hexahistidine [His$_6$] tag), the central domain of *B. saltans* KKT2 (TriTrypDB accession no. BSAL_50690) was amplified from BAG50 (a synthetic DNA that encodes *B. saltans* KKT2, codon optimized for expression in Sf9 insect cells; Table S4) with primers BA2117/BA2118 and cloned into the RSFDuet-1 vector (Novagen) using BamHI/EcoRI sites with an NEBuilder HiFi DNA Assembly Cloning Kit (New England Biolabs) according to the manufacturer's instructions. To make pBA1139 (His$_6$-$PkKKT2^{551-679}$), the central domain of *Perkinsela* CCAP 1560/4 KKT2a (UniProt accession no. XU18_4017) was amplified from BAG48 (a synthetic DNA that encodes *Perkinsela* KKT2a, codon optimized for expression in Sf9 insect cells; Table S4) with primers BA1569/BA1570 and cloned into RSFDuet-1 using BamHI/EcoRI sites with an In-Fusion HD Cloning Plus kit (Takara). To make pBA2276 (His$_6$-designed zinc finger), designed zinc finger domain was amplified from BAG136 (a synthetic DNA codon optimized for expression in *E. coli*) with primers BA3077/BA3078 and cloned into RSFDuet-1 using BamHI/EcoRI sites with an NEBuilder HiFi DNA Assembly Cloning Kit. To make pBA283 (His$_6$-$TbKKT2^{562-677}$), the central domain of *T. brucei* KKT2 (TriTrypDB accession no. Tb927.11.10520) was amplified from genomic DNA with primers BA670/BA574 and cloned into pNIC28-Bsa4 using ligation-independent cloning. To make pBA1178 (His$_6$-$TbKKT2^{562-677}$ C597A; C600A), the central domain of *T. brucei* KKT2 was amplified from pBA493 (a plasmid that encodes full-length *T. brucei* KKT2 harboring C597A; C600A mutations) with primers BA1563/BA1567 and cloned into RSFDuet-1 using BamHI/EcoRI sites with an NEBuilder HiFi DNA Assembly Cloning Kit. Recombinant proteins were expressed in BL21(DE3) *E. coli* cells at 20°C using auto induction media (Formedium; Studier, 2005).

Briefly, 500 ml of cells were grown at 37°C in 2.5-liter flasks at 300 rpm until OD$_{600}$ of 0.2–0.3 and then cooled down to 20°C overnight (2 liters for *Bs*KKT2, 6 liters for *Pk*KKT2a, 2 liters for designed zinc finger, 6 liters for *Tb*KKT2, and 12 liters for *Tb*KKT2 C597A; C600A). Cells were harvested by centrifugation and resuspended in 50 ml per liter of culture of lysis buffer (25 mM Hepes, pH 7.5, 150 mM NaCl, 1 mM tris(2-carboxyethyl) phosphine (TCEP), 10 mM imidazole, and 1.2 mM PMSF). Proteins were extracted by mechanical cell disruption using a French press (1 passage at 20,000 PSI), and the resulting lysate was centrifuged at 48,384 *g* for 30 min at 4°C. Clarified lysate was incubated with 5 ml TALON beads (Takara), washed with 150 ml lysis buffer, and eluted in 22 ml of elution buffer (25 mM Hepes, pH 7.5, 150 mM NaCl, 1 mM TCEP, and 250 mM imidazole) in a gravity column, followed by tobacco etch virus treatment for the removal of the His$_6$ tag. Salt concentration of the sample was subsequently reduced to 50 mM NaCl using buffer A (50 mM Hepes, pH 7.5, and 1 mM TCEP) and the sample was loaded onto a 5-ml HiTrap Heparin HP affinity column (GE Healthcare) preequilibrated with 5% buffer B (50 mM Hepes, pH 7.5, 1 M NaCl, and 1 mM TCEP) on an ÄKTA pure 25 system. Protein was eluted by using a gradient from 0.05 to 1 M NaCl, and protein-containing fractions were combined, concentrated with an Amicon stirred cell using an ultrafiltration disc with 10-kD cutoff (Merck), and then loaded onto a HiPrep Superdex 75 16/60 size exclusion chromatography column (GE Healthcare) preequilibrated with 25 mM Hepes (pH 7.5), 150 mM NaCl, and 1 mM TCEP. Fractions containing the protein of interest were pooled together, concentrated with an Amicon stirred cell using an ultrafiltration disc with 10-kD cutoff, and stored at –80°C. Designed zinc finger was buffered exchanged into 50 mM Tris (pH 7.5), 1 mM ZnCl$_2$, 50 mM NaCl, and 1 mM TCEP prior to storage. Protein concentration was measured by Bradford assay.

## Crystallization

Both $BsKKT2^{572-668}$ and $PkKKT2a^{551-679}$ crystals were optimized at 4°C in sitting drop vapor diffusion experiments in 48-well plates using drops of overall volume 400 nl, mixing protein, and mother liquor in a 3:1 protein:mother liquor ratio. *Bs*KKT2 central domain crystals grew from the protein at 26 mg/ml and mother liquor 40% PEG 400, 0.2 mM (NH$_4$)$_2$SO$_4$, and 100 mM Tris-HCl (pH 8). The 40% PEG400 in the mother liquor served as the cryoprotectant when flash-cooling the crystals by plunging into liquid nitrogen. $PkKKT2a^{551-679}$ crystals grew from the protein at 13 mg/ml and mother liquor 19% 2-methyl-2,4-pentanediol (MPD), 50 mM Hepes, pH 7.5, and 10 mM MgCl$_2$. The crystals were briefly transferred into a cryoprotecting solution of 30% MPD, 50 mM Hepes, pH 7.5, and 10 mM MgCl$_2$ before flash cooling.

## Data collection and structure determination

X-ray diffraction data from a *Bs*KKT2 central domain crystal were collected at the I04 beamline at the Diamond Light Source at the Zinc K-edge wavelength ($\lambda$ = 1.28297 Å). A set of 1,441 images were processed in space group I222 using the Xia2 pipeline (Winter, 2010), with DIALS for indexing and integration (Winter et al., 2018) and AIMLESS for scaling (Evans and Murshudov, 2013) to 1.8-Å resolution. Three initial Zn atoms were localized by interpreting the anomalous difference Patterson, SAD phases were estimated using Crank2 (Skubák and Pannu, 2013), and an initial model was built with BUCCANEER (Cowtan, 2006). The structure was completed by several cycles of alternating model building in Coot (Emsley et al., 2010) and

refinement in autoBUSTER (Blanc et al., 2004; Bricogne et al., 2017).

*Pk*KKT2a$^{551-679}$ x-ray diffraction data were collected at the I03 beamline at Diamond Light Source also at the zinc K-edge ($\lambda$ = 1.28272 Å) and processed using the autoPROC pipeline (Vonrhein et al., 2011) using XDS (Kabsch, 2010) for indexing/integration and AIMLESS (Evans and Murshudov, 2013) for scaling to a resolution of 3.8 Å. Two initial Zn positions were determined by interpreting the anomalous difference Patterson, and SAD phases were estimated using Crank2 (Skubák and Pannu, 2013) and SHARP (Vonrhein et al., 2007) in space group P6$_4$. An initial model was manually built in Coot and refined once with RosettaMR (Terwilliger et al., 2012). The structure was completed by several cycles of alternating model building in Coot (Emsley et al., 2010) and refinement in autoBUSTER (Blanc et al., 2004; Bricogne et al., 2017).

A higher-resolution dataset was collected from a *Pk*KKT2a$^{551-679}$ crystal at the I24 beamline at Diamond Light Source at a wavelength of $\lambda$ = 0.9686 Å. Data were processed using Xia2 pipeline (Winter, 2010), DIALS (Winter et al., 2018), and AIMLESS (Evans and Murshudov, 2013) in space group P6$_4$ to a resolution of 2.9 Å. The model obtained from the 3.8-Å dataset was used for further model building and refinement with autoBuster (Bricogne et al., 2017) and Coot (Emsley et al., 2010).

All images were made with Pymol (Schrödinger) and CCP4mg (McNicholas et al., 2011). Topology diagrams were generated using TopDraw (Bond, 2003). Protein coordinates have been deposited in the RCSB Protein Data Bank (http://www.rcsb.org/) with accession nos. 6TLY (*B. saltans* KKT2) and 6TLX (*Perkinsela* KKT2a).

### EMSA
DNA oligonucleotides were purchased from Thermo Fisher Scientific (Table S4). To make double-stranded DNA (dsDNA), 100-μM stock solutions were mixed at 1:1 vol/vol ratio and incubated on a thermo block at 95°C for 2 min followed by cooling by dissipation to RT. Prior to the assay, proteins were buffer exchanged into binding buffer (25 mM Hepes, pH 7.5, 50 mM NaCl, and 1 mM TCEP) using a Zeba spin desalting column (Thermo Fisher Scientific), serially diluted at 2:3 vol/vol ratio (30 μM, 20 μM, 13.3 μM, 8.9 μM, 5.9 μM, 3.9 μM, 2.6 μM, 1.8 μM, 1.2 μM, and 0.78 μM), and then incubated with 200 nM final concentration of dsDNA probes for 30 min at RT. Before each experiment, native-PAGE gels (NativePAGE, Bis-Tris 4–16%; Thermo Fisher Scientific) were prerun at 150 V for 20 min at 4°C. For each experiment, 10-μl samples were loaded onto native-PAGE gels and run without cathode buffer at 100 V for 1 h at 4°C. 100-bp DNA ladder was loaded as a marker (N3231L; New England Biolabs). Gels were stained with GelRed (Biotium) and imaged using Gel Doc XR+ (Bio-Rad).

### Fluorescence anisotropy DNA-binding assay
All experiments were performed in binding buffer (25 mM Hepes, pH 7.5, 50 mM NaCl, and 1 mM TCEP) using 1 nM fluorescein amidite–labeled dsDNA sequences purchased from IDT (Table S4). Prior to the assay, proteins were buffer exchanged into binding buffer using a Zeba spin desalting column (Thermo Fisher Scientific), serially diluted at 2:3 vol:vol ratio, and then incubated with DNA for 20 min at RT. Fluorescence anisotropy was measured at 25°C using a PHERAstar FS next-generation microplate reader (BMG-Labtech). Each data point is an average of three independent experiments. Data were fitted with SigmaPlot using a standard four-parameter logistic equation to calculate $K_d$.

### Online supplemental material
Fig. S1 illustrates that KKT2 fragments localize at kinetochores from S phase to anaphase, while the KKT3 central domain localizes constitutively. Fig. S2 shows purification of recombinant proteins used in this study. Fig. S3 demonstrates multiple sequence alignment of KKT2 and KKT3 homologues, highlighting the strict conservation of an aspartic acid residue. Fig. S4 shows DNA-binding assays for *Pk*KKT2 and *Tb*KKT2 central domains. Fig. S5 shows analysis of KKT2 and KKT3 mutants in trypanosomes. Table S1 lists the proteins identified in the immunoprecipitates of YFP-tagged KKT2$^{558-679}$, KKT2$^{672-1030}$, KKT2$^{1024-1260}$, KKT2$^{1024-1260\ (W1048A)}$, KKT3$^{594-811}$, and KKT3$^{846-1058}$ by mass spectrometry. Table S2 lists the DALI search hits for the *Bs*KKT2 CL domain structure. Table S3 lists the DALI search hits for the *Bs*KKT2 C2H2 zinc finger structure. Table S4 lists the trypanosome cell lines, plasmids, primers, synthetic DNA, and DNA probes used in this study.

### Data availability
Protein coordinates have been deposited in the RCSB Protein Data Bank (http://www.rcsb.org/) with accession nos. 6TLY (*B. saltans* KKT2) and 6TLX (*Perkinsela* KKT2a).

## Acknowledgments
We thank Pietro Roversi and Matt Higgins for advice, crystallography facility manager Edward Lowe, Micron Advanced Bioimaging Unit, and Advanced Proteomics Facility. We also thank Pietro Roversi, Matt Higgins, Danny Huang, and Patryk Ludzia for comments on the manuscript.

M. Ishii was supported by a long-term fellowship from the TOYOBO Biotechnology Foundation. B. Akiyoshi was supported by a Wellcome Trust Senior Research Fellowship (grant 210622/Z/18/Z) and the European Molecular Biology Organization Young Investigator Program.

The authors declare no competing financial interests.

Author contributions: G. Marcianò purified recombinant proteins, solved crystal structures, and performed DNA-binding assays. M. Ishii performed RNAi and rescue experiments. O.O. Nerusheva expressed KKT2 and KKT3 truncations and mutants in trypanosomes and performed immunoprecipitation and mass spectrometry. B. Akiyoshi performed RNAi experiments and expressed KKT2 and KKT3 truncations and mutants in trypanosomes. G. Marcianò, M. Ishii, and B. Akiyoshi wrote the manuscript.

Submitted: 5 January 2021

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

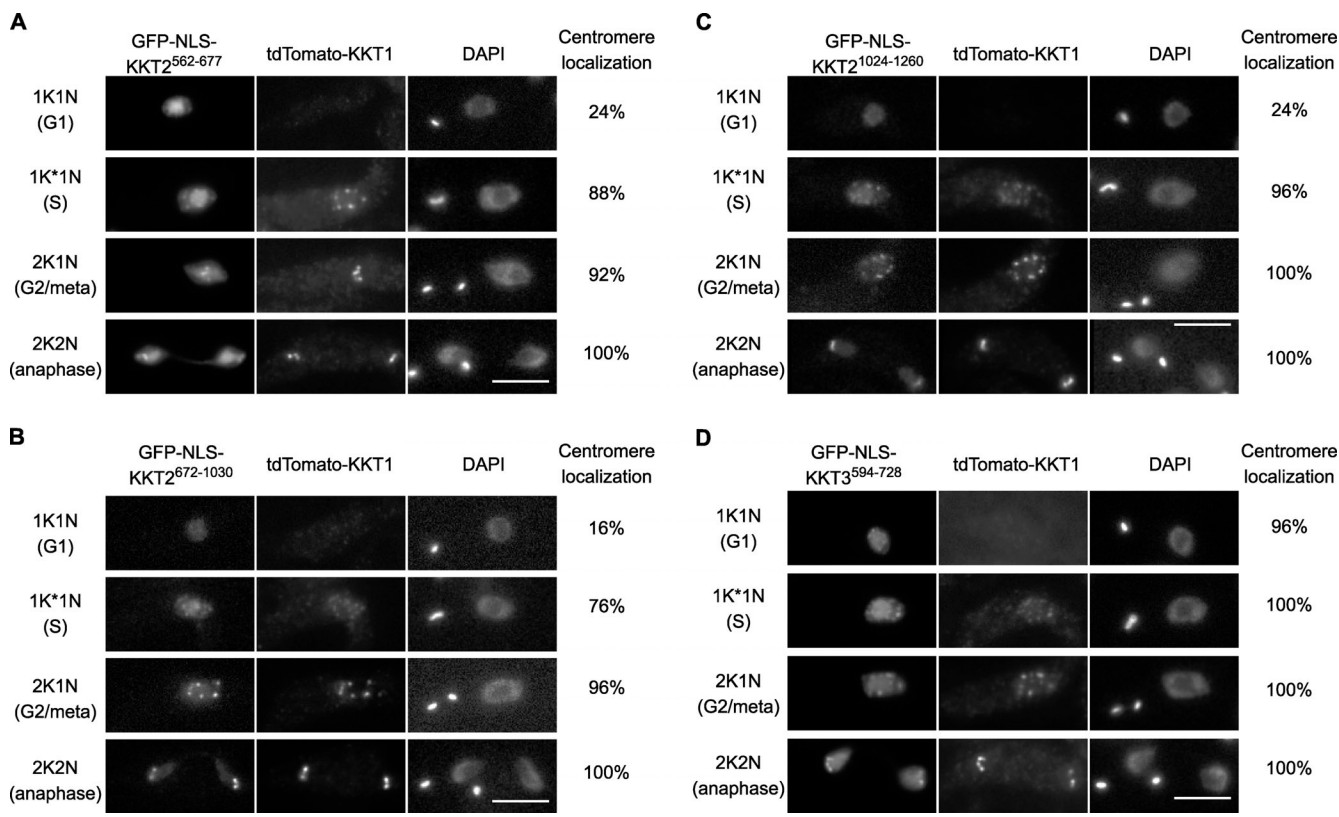

Figure S1. **KKT2 fragments localize at kinetochores from S phase to anaphase, while KKT3 central domain localizes constitutively. (A–C)** Ectopically expressed *Tb*KKT2 central domain (562–677), *Tb*KKT2$^{672-1030}$, and *Tb*KKT2 DPB (1,024–1,260) localize at kinetochores from S phase until anaphase. **(D)** Ectopically expressed *Tb*KKT3 central domain (594–728) forms kinetochore-like dots throughout the cell cycle. Inducible GFP-NLS fusion proteins were expressed with 10 ng/ml doxycycline. tdTomato-KKT1 was used as a kinetochore marker (*n* = 25 in each cell cycle stage). Cell lines: BAP1998, BAP2000, BAP1999, and BAP1997. Scale bars, 5 µm.

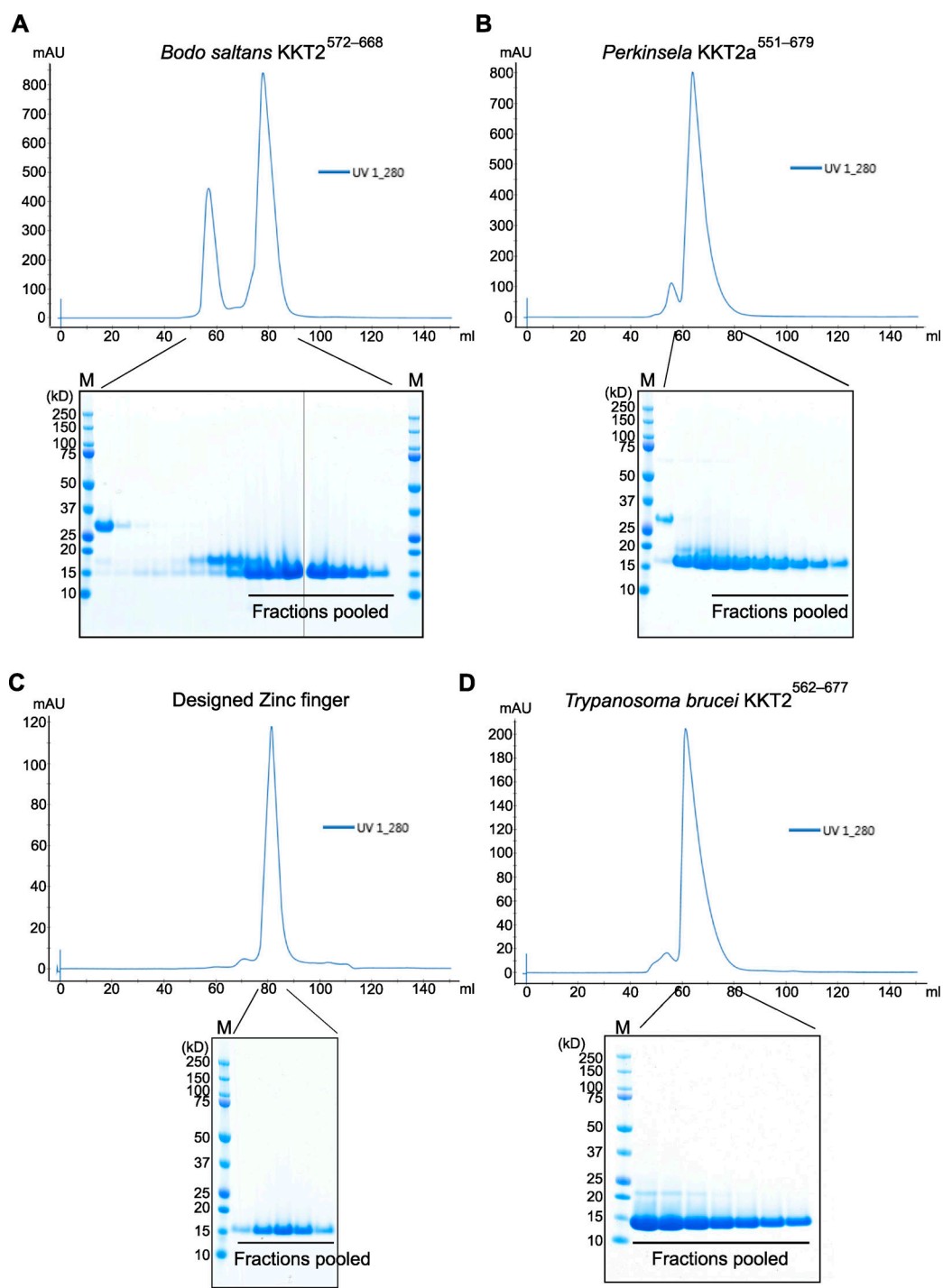

Figure S2. **Purification of *B. saltans* KKT2, *Perkinsela* KKT2a, designed zinc finger, and *T. brucei* KKT2 proteins. (A–D)** Size exclusion chromatography of *Bs*KKT2 central domain (A), *Pk*KKT2a$^{551–679}$ (B), designed zinc finger (C), and *Tb*KKT2 central domain (D) with respective SDS-PAGE gels showing pooled fractions. HiPrep Superdex 75 16/60 column was used. Note that two separate gels are shown for A. M, protein marker.

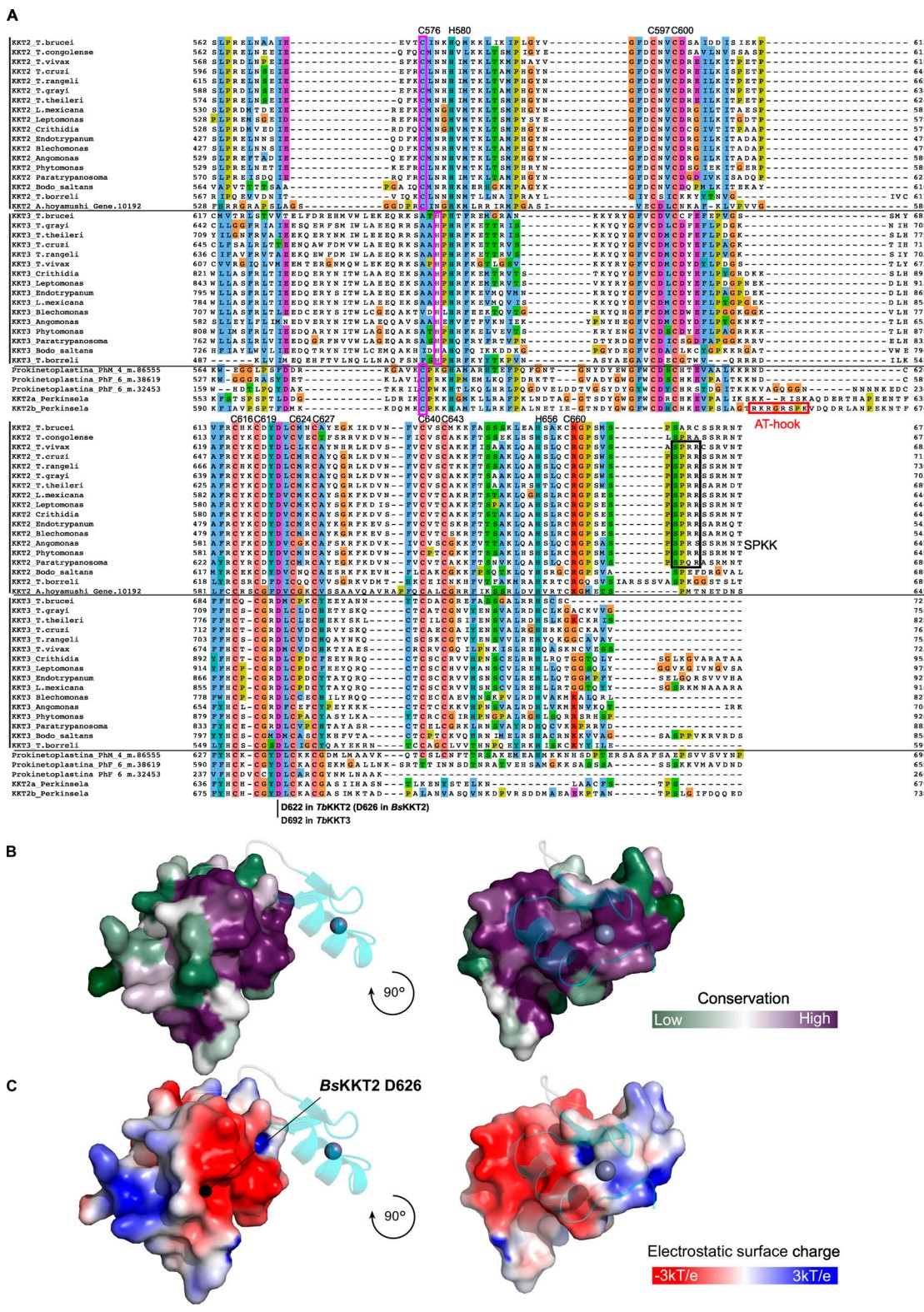

Figure S3. **Multiple sequence alignment of KKT2 and KKT3 homologues in kinetoplastids reveals a conserved aspartic acid residue (D622 in *Tb*KKT2, D626 in *Bs*KKT2). (A)** Residue numbers in *Tb*KKT2 for those cysteines and histidines that coordinate zinc ions in the *Bs*KKT2 structure are listed at the top of the alignment to highlight the conservation of these residues among kinetoplastids. Note that cysteine is used in KKT2 and KKT2-like proteins (C576 in *Tb*KKT2), while histidine is present in KKT3 in slightly different position (highlighted in pink box). The position of the strictly conserved aspartic acid residue is also shown. SPKK motifs (black box) and a putative AT-hook motif in *Perkinsela* KKT2b (red box) are also highlighted. **(B)** Surface sequence conservation of *Bs*KKT2 CL domain using the ConSurf server (Landau et al., 2005; Ashkenazy et al., 2016). The C2H2 zinc finger structure is shown as a cartoon representation. **(C)** Electrostatic surface potential of the *Bs*KKT2 CL domain generated by Adaptive Poisson–Boltzmann Solver (Jurrus et al., 2018) reveals the presence of a conserved acidic surface. The location of the residue D626 is marked by a black circle.

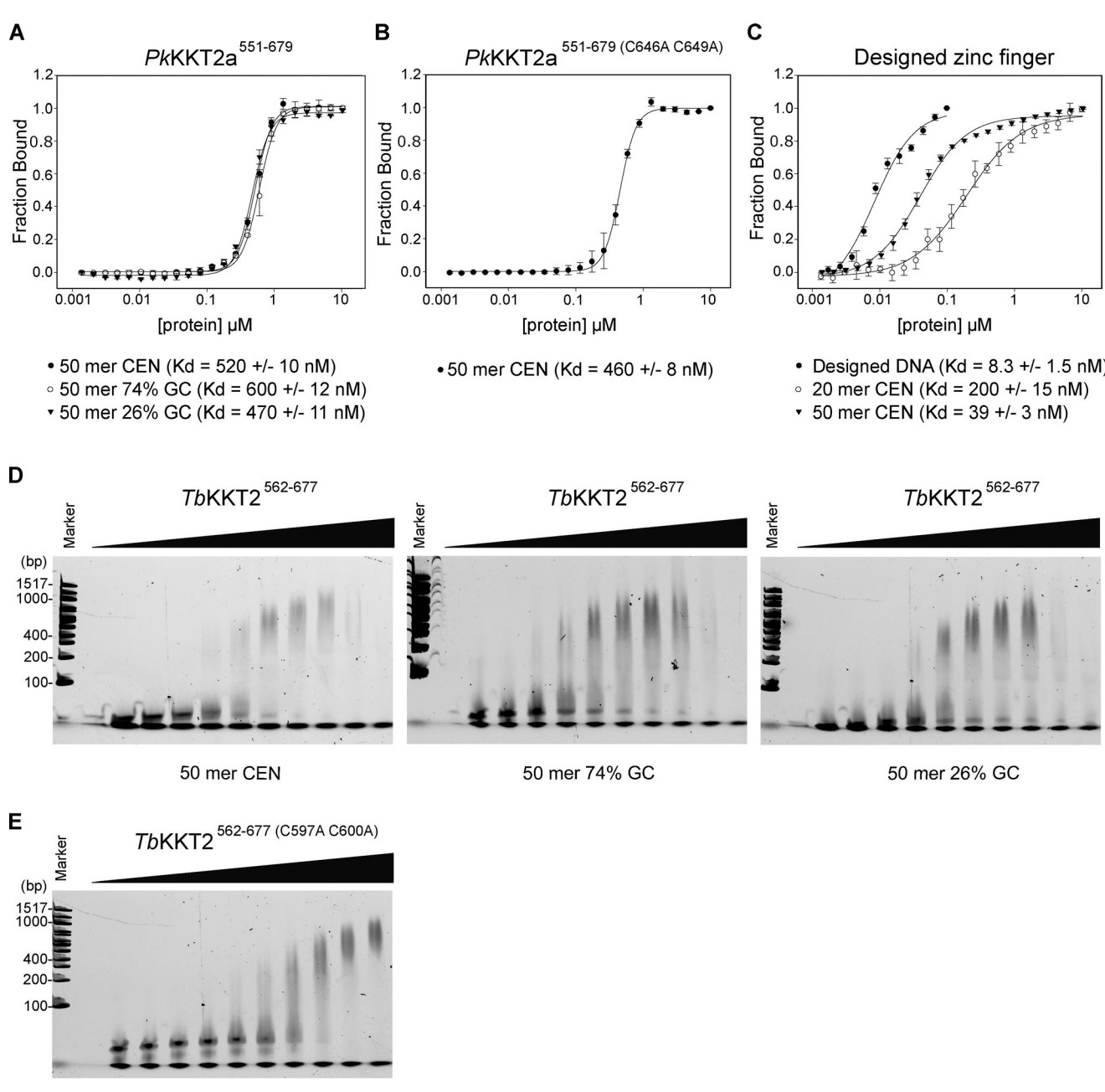

Figure S4. **KKT2 central domain has weak DNA binding. (A)** Fluorescence polarization assay for *Pk*KKT2a[551–679] on 50-bp DNA probes of different GC contents, showing that it binds DNA in a sequence-independent manner. 50 mer CEN probe is part of the centromeric sequence (CIR147) in *T. brucei* and has 36% GC content. Probes: BA1674, BA2218, and BA2216. **(B)** Fluorescence polarization assay for *Pk*KKT2a[551–679 (C646 C649A)], showing that the mutant has similar DNA-binding affinity. Probe: BA1674. **(C)** Fluorescence polarization assay for designed zinc finger (Jantz and Berg, 2010), showing that it has sequence-specific DNA-binding activity (designed DNA is 25 bp). 20 mer CEN DNA has 35% GC content. Probes: BA3083, BA1793, and BA1674. **(D)** EMSA for *Tb*KKT2[562-677] (proteins were serially diluted 1.5-fold starting from 30 μM: 30 μM, 20 μM, 13.3 μM, 8.9 μM, 5.9 μM, 3.9 μM, 2.6 μM, 1.8 μM, 1.2 μM, and 0.78 μM) on 50-bp DNA probes of different GC contents, showing that it binds DNA in a sequence-independent manner. 50 mer CEN probe is part of the centromeric sequence (CIR147) in *T. brucei* and has 36% GC content. Probes: BA3296/BA3297 (CEN), BA3300/BA3301 (74% GC), and BA3298/BA3299 (26% GC). **(E)** EMSA for *Tb*KKT2a[562–677 (C597 C600A)] (from 30 μM to 0.78 μM as above), showing that the mutant has a similar DNA-binding affinity as WT. Probe: BA3296/BA3297 (CEN).

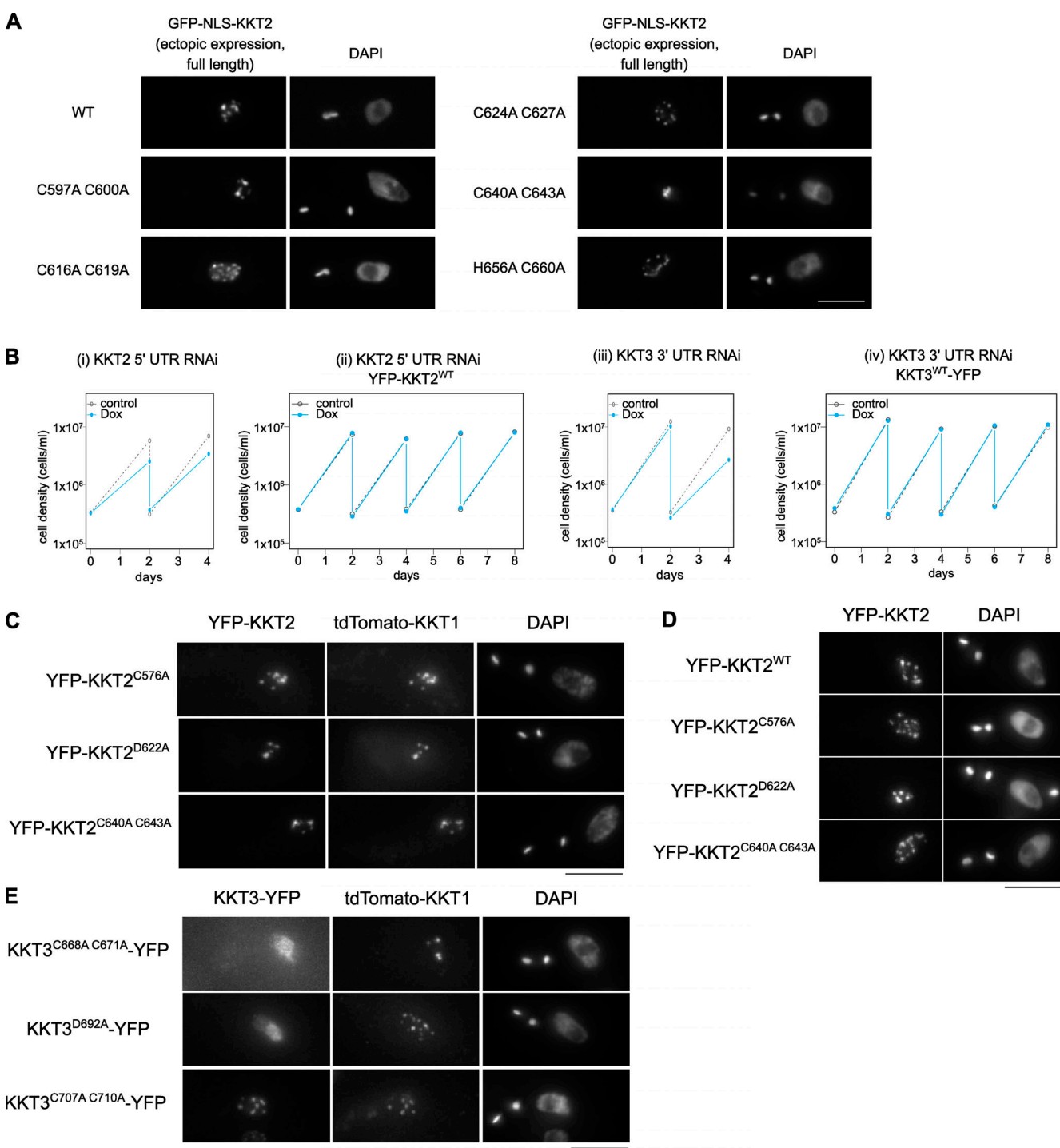

Figure S5.  **Analysis of KKT2 and KKT3 mutants in trypanosomes. (A)** Full-length KKT2 proteins with indicated mutations were ectopically expressed with 10 ng/ml doxycycline (Dox) for 1 d. Cell lines: BAP327, BAP365, BAP366, BAP367, BAP368, and BAP369. Scale bar, 5 µm. **(B)** RNAi of KKT2 and KKT3 causes growth defects in *T. brucei*. Growth curves of KKT2 5′UTR RNAi (i), KKT2 5′UTR RNAi with YFP-KKT2 (resistant to the RNAi; ii), KKT3 3′UTR RNAi (iii), and KKT3 3′UTR RNAi with KKT3-YFP (resistant to the RNAi; iv). 1 µg/ml doxycycline was added to induce RNAi. Controls are uninduced cell cultures. Similar results were obtained from at least two independent experiments. Cell lines: BAP1554, BAP1681, BAP1555, and BAP1682. **(C)** Indicated KKT2 mutants colocalize with a kinetochore marker, tdTomato-KKT1. Cell lines: BAP2036, BAP2033, and BAP2035. **(D)** KKT2 CL domain mutants localize at kinetochores even when endogenous KKT2 protein is depleted (*n* > 50, 2K1N cells). One allele of KKT2 was mutated and tagged with an N-terminal YFP tag, and the other allele was depleted for 4 d using RNAi-mediated knockdown by targeting the 5′UTR of the KKT2 transcript. Maximum intensity projections are shown. RNAi was induced with 1 µg/ml doxycycline. Cell lines: BAP1681, BAP1789, BAP1779, and BAP1786. **(E)** KKT3$^{C707A\ C710A}$ colocalizes with tdTomato-KKT1, while KKT3$^{C668A\ C671A}$ and KKT3$^{D692A}$ mutants do not. Cell lines: BAP2037, BAP2038, and BAP2034. Scale bars, 5 µm.

Four tables are provided online. Table S1 lists the proteins identified by mass spectrometry in the immunoprecipitates of YFP-tagged KKT2$^{558-679}$, KKT2$^{672-1030}$, KKT2$^{1024-1260}$, KKT2$^{1024-1260\ (W1048A)}$, KKT3$^{594-811}$, and KKT3$^{846-1058}$. Table S2 lists the DALI search hits for the *Bs*KKT2 CL domain structure. Table S3 lists the DALI search hits for the *Bs*KKT2 C2H2 zinc finger structure. Table S4 lists the trypanosome cell lines, plasmids, primers, synthetic DNA, and DNA probes used in this study.

