## [Peer Review File · The Journal of Cell Biology]

Kinetoplastid kinetochore proteins KKT2 and KKT3 have unique centromere localization domains

Gabriele Marcianò, Midori Ishii, Olga Nerusheva, and Bungo Akiyoshi

Corresponding Author(s): Bungo Akiyoshi, University of Oxford

Review Timeline:

Submission Date:	2021-01-05
Editorial Decision:	2021-02-12
Revision Received:	2021-04-29
Editorial Decision:	2021-05-10
Revision Received:	2021-05-11

Monitoring Editor: Arshad Desai

Scientific Editor: Dan Simon

Transaction Report:

DOI: <https://doi.org/10.1083/jcb.202101022>

February 12, 2021

Re: JCB manuscript #202101022

Dr. Bungo Akiyoshi
University of Oxford
South Parks Road
Oxford OX1 3QU
United Kingdom

Dear Bungo,

Thank you for submitting your manuscript "Unconventional kinetochore kinases KKT2 and KKT3 have unique centromere localization domains" for consideration to the Journal of Cell Biology. The manuscript has been evaluated by three expert reviewers, whose feedback is appended to this letter. Based on their comments, we would like to invite you to submit a revision.

You will see that the reviewers are supportive of the work but raise a number of points that we believe are important to address. In particular, there needs to be greater clarity on the relationship between the localization and DNA binding activity of the CL domain. While the *T. brucei* proteins did not work in the polarization-based assay, their DNA-binding activity should be tested using alternative approaches, such as gel shift/EMSA. Such an effort would help relate mutations generated in the central domain and their effects on localization to potential DNA binding activity. Reviewers also request clarification on a number of issues, additional controls and have important suggestions on points to cover in the text, all of which should be addressed in the revision. Finally, Reviewer #1 asks whether the kinase activities of KKT2 & KKT3 play a role in their kinetochore localization/functions and Reviewer #3 asks whether the Polo box domain resembles canonical phospho-peptide binding Polo box domains. Addressing these points would enhance the impact of the work, and we would encourage you to consider adding them into the revision if possible. However, we will not require these to be experimentally addressed but would recommend at the least that they be discussed in the text, as these points are very likely to occur to an interested reader based on the primary structures of KKT2 and KKT3.

GENERAL GUIDELINES:

Text limits: Character count for an Article is < 40,000, not including spaces. Count includes title page, abstract, introduction, results, discussion, acknowledgments, and figure legends. Count does not include materials and methods, references, tables, or supplemental legends.

Figures: Articles may have up to 10 main text figures. Figures must be prepared according to the policies outlined in our Instructions to Authors, under Data Presentation, <https://jcb.rupress.org/site/misc/ifora.xhtml>. All figures in accepted manuscripts will be screened prior to publication.

IMPORTANT: It is JCB policy that if requested, original data images must be made available. Failure to provide original images upon request will result in unavoidable delays in publication. Please ensure that you have access to all original microscopy and blot data images before submitting your revision.

Supplemental information: There are strict limits on the allowable amount of supplemental data. Articles may have up to 5 supplemental figures. Up to 10 supplemental videos or flash animations are allowed. A summary of all supplemental material should appear at the end of the Materials and methods section.

As you may know, the typical timeframe for revisions is three to four months. However, we at JCB realize that the implementation of social distancing and shelter in place measures that limit spread of COVID-19 also pose challenges to scientific researchers. Lab closures especially are preventing scientists from conducting experiments to further their research. Therefore, JCB has waived the revision time limit. We recommend that you reach out to the editors once your lab has reopened to decide on an appropriate time frame for resubmission. Please note that papers are generally considered through only one revision cycle, so any revised manuscript will likely be either accepted or rejected.

Thank you for this interesting contribution to Journal of Cell Biology. You can contact us at the journal office with any questions, cellbio@rockefeller.edu or call (212) 327-8588.

Sincerely,

Arshad Desai, Ph.D.
Monitoring Editor
Journal of Cell Biology

Dan Simon, Ph.D.
Scientific Editor
Journal of Cell Biology

Reviewer #1 (Comments to the Authors (Required)):

In most eukaryotic cells, kinetochore assembly relies on centromere-specific CENP-A nucleosomes and constitutive centromere-associated network proteins (such as CENP-C). However, kinetoplastids including *Trypanosoma* do not have such canonical kinetochore proteins. The authors' group and others have recently identified dozens of non-canonical kinetochore proteins in kinetoplastids. However, it is still unclear which proteins directly bind centromere DNA to establish foundation of kinetochore assembly.

This study has found that KKT2 and KKT3 are DNA-binding proteins, which constitutively localize at the centromere/kinetochore and play important roles in assembly of several other kinetochore proteins. On the other hand, localization of KKT2 and KKT3 are not affected by depletion of other kinetochore proteins. Therefore, it is likely that KKT2 and KKT3 directly bind centromere DNA and establish foundation of kinetochore assembly. Moreover, this study identified and characterized DNA binding motifs of KKT2 and KKT3 through structural studies and mutation analyses.

Most experiments were carried out in high standard and the results generally support the authors' conclusions. This work gives important insights into how centromeres are recognized to promote assembly of non-canonical kinetochores in kinetoplastids. I support publication of this manuscript in JCB, however authors need to strengthen some data before publication, as follows:

Specific points:

1) Figure 1: In Figure 1C, KKT3-YFP seems to be significantly reduced after KKT8 and KKT22 are depleted, which may not be consistent with their conclusion in text. Quantitative data should be shown to compare KKT2-YFP and KKT3-YFP localization with control vs various RNAi (e.g. percentage of cells showing YFP signals at kinetochores).

2) Figure 2B, E, H: I wonder what fraction of 2K1N cells show defective localization of YFP signals with control RNAi. Is it always 0%? If so, please clarify it in the figure legend. If not, results with control RNAi must be included in figures.

3) Figure 7D and E: These results show that C646A C649A mutations do not change DNA binding affinity of PkKKT2 in vitro. However, corresponding mutations C616A and C619A abolish the centromere/kinetochore localization of TbKKT2 in vivo (Figure 8B). I wonder how the authors explain this discrepancy.

Optional point:

KKT2 and KKT3 are unique kinases that constitutively localize at centromeres/kinetochores and promote assembly of several other kinetochore proteins. I wonder if it is possible to address whether their kinase activity is required for such functions. For example, to address this, they can test whether kinase-negative forms of KKT2 and KKT3 rescue the defects caused by depletion of original KKT2 and KKT3. I suggest this experiment as optional, in case it is technically difficult or it requires considerable amount of time to complete.

Reviewer #2 (Comments to the Authors (Required)):

This study focusses on the characterization of two kinetochore factors, KKT2 and KKT3 and their contribution to the assembly of the unconventional kinetochore complex in Kinetoplastids. These organisms lack components of canonical kinetochore complexes including the essential kinetochore initiation factor CENP-A. The assembly and organization of kinetoplastid kinetochores is largely unknown with some important insights revealed in previous studies mainly from this group. KKT2 and KKT3 are interesting to analyze in this context because these components (among some others) localize constitutively to centromeres throughout the cell cycle - a feature associated with

factors that act at the base for kinetochore assembly.

To get more insights into kinetochore assembly, the authors evaluated the reciprocal relationship between KKT2 and 3 (and both) and other kinetochore components for their recruitment using RNAi-mediated depletions and visualization of fluorescence-tagged factors in *T. brucei* as their model. By expressing KKT2 or KKT3 protein fragments, they then continued to dissect the contribution of individual domains for centromere localization and kinetochore protein interaction. These analyses revealed that in particular the C-terminal domains of KKT2 and 3 interact with multiple other kinetochore proteins. The authors thus hypothesize that KKT2 and KKT3 might be important for the recruitment of other kinetochore components (or vice versa). The central domains of KKT2 and 3 interact with fewer kinetochore components but appear to be able to localize to centromeres.

Turning to other kinetoplastids due to technical reasons, they identified two Zn-finger binding domains in the central domain of *B. saltans* using structural analyses. One of the two, to which they refer to as CL domain is also found in the more distant kinetoplastid *Perkinsella*. Following up on the *Perkinsella* KKT2a central domain they further found that it has DNA binding activity, though with relatively low affinity. Interestingly, an intact structure of the protein does not appear to be important for this activity. Returning to *T. brucei* for functional analyses, they found that residues involved in Zn coordination and acidic sites within the CL domain are important for centromere localization of KKT2 and KKT3 central domains and cell viability.

This is an interesting study shedding new light on the contribution of proteins involved in potentially early steps of kinetochore assembly in kinetoplastids. I recommend the study for publication in JCB after addressing the following minor issues and questions.

The results showing unaffected KKT2 and KKT3 localization in cells depleted for other KKT proteins is convincing. If I understand correctly, while they could only achieve efficient depletion of the kinetochore protein of interest in about 60% of cells (leaving 40% of cells in which depletion is incomplete), KKT2 and KKT3 signal was not observed in more than 94 or 97% of cells. The data on the effect of KKT2 and 3 depletion on the recruitment of other kinetochore components are however weaker, probably due to technical reasons related to inefficient depletions of these proteins (as also stated in the text and the figure legend). For Figure 2H, data of only a subset of kinetochore components is displayed. To enable a better comparison with the single depletions, it would be good if the same kinetochore components as those shown in B and E would be analyzed in this panel as well.

In general, it would also be helpful if the authors specify how they determined absence of a kinetochore protein. Was a certain threshold of intensity used to conclude presence or absence? If data on signal intensities are available, I think it would contribute to the study if those were displayed to discuss potential quantitative effects on kinetochore protein localization.

There is a bit of a discrepancy between the identity of kinetochore components with affected localization upon KKT2 or KKT3 depletion and those found in their MS data. Of course, kinetochore components with affected centromere localization might not be direct protein binding partners of KKT2 or KKT3 and thus not identified in their proteomic datasets. One could have however expected that the localization of protein binding partners would be affected upon KKT2 or KKT3 depletions. If the authors agree, I think it would be good if they comment on this aspect.

Proteins identified in KKT2(558-679) IP-MS: Are peptides in Table of panel C mapping to the full-length or only the expressed fragment of KKT2? If the authors recovered the full-length KKT2 protein, could the recruitment of the fragment also be due to dimerization of the protein in this part

instead of DNA binding? Same for KKT3, Figure 4E.

Figure 8. In contrast to KKT3-CL mutants that interestingly are completely unable to localize to centromeres, KKT2 CL mutants can at least at time 0 prior to RNAi induction against the endogenous copy, probably due to the interaction with other kinetochore proteins as hypothesized by the authors. It would be interesting to test whether KKT2 CL mutant localization is impaired at later time points when the growth defects arise. This would suggest that kinetochore complex might indeed be destabilized in the presence of these mutations.

A model for kinetochore assembly summarizing the findings of this paper and the current knowledge would be very helpful include at the end of the paper.

I think there is a mix-up between the links for the supplemental tables. Link to Table S1 opens Table S3 for example...

Reviewer #3 (Comments to the Authors (Required)):

Kinetoplastids are evolutionarily divergent eukaryotes with no detectable equivalent of CENP-A, a Histone H3 variant that defines centromeres in most eukaryotes including yeast and humans or any other canonical kinetochore proteins. How the kinetoplastid centromere is defined, what constitutes the kinetochore and how functional kinetochore is assembled are important questions yet to be addressed at a molecular level.

This work from Marciano & Ishii et al., provides insights into the molecular determinants of kinetochore localisation of KKT2 and KKT3, two constitutive members of the kinetoplastid kinetochore. Both KKT2 and KKT3 possess an N-terminal kinase domain, C-terminal polo-box like domain (DPB) and an uncharacterised but conserved central domain. Using RNAi based depletion experiments, they show that: 1) KKT2 and KKT3 can localise to kinetochores independently of most of the other known kinetochore components, 2) kinetochore localisation of KKT14 and KKT23/KKT24 depend on KKT2 and KKT3, respectively. Localisation studies performed using various short fragments of KKT2 and KKT3 showed the central domain is crucial for their kinetochore association. IP/MS carried out for the central and polo-box like domains of KKT2 and KKT3 co-purified several other KKT proteins. Crystal structure analyses of central domains of *Bodo saltans* KKT2 and *Perkinsela* KKT2a (KKT2-like) revealed the presence of two distinct zinc-binding domains which share structural similarity to atypical C1 domain of Vav1 protein. Either disrupting the central domain integrity by mutating the Zn-coordinating cysteines or disrupting a conserved negatively charged surface patch by mutating a highly conserved Asp acid abolished KKT2 and KKT3's ability to localise at centromeres and failed to rescue the growth defect observed upon depletion of wt proteins. Overall this work highlights the requirement of the central domains of KKT2 and KKT3 for their centromere association.

Points that need to be addressed:

How the central domains of KKT2 and KKT3 contribute to their centromere localisation is still not clear. The authors propose that the central domain of KKT2 and KKT3 might have DNA-binding activity. Due to technical difficulty the authors could not test this hypothesis for KKT2 and KKT3, but instead they show that Pk KKT2a CL-like domain can indeed bind DNA. In my opinion, testing the DNA binding ability of the central domains of KKT2 and KKT3 by EMSA (Electrophoretic Mobility

Shift Assay) or ITC (Isothermal Titration Calorimetry) will improve the impact of this work and needs to be tested.

Surprisingly, disrupting the integrity of the PkKKT2a CL domain by mutating the Zn coordinating cys residues did not affect its ability to bind DNA. The authors do not really comment about this intriguing observation. This clarification is important as equivalent mutations in Tb KKT2 and KKT3 (disrupting the integrity of CL domain) affect their centromere localisation.

Mutating the conserved Asp residue of the negatively charged surface (in KKT2 CL and KKT3 CL) affects centromere localisation of KKT2 and KKT3. What I find intriguing is that a subtle variation at this position, ASP to GLU (which is unlikely to change the electrostatic surface charge of the patch), is sufficient to disrupt the centromere localisation. I would like to see the authors explaining the rationale for mutating this residue to GLU and comment on the observation. It would certainly help if authors could test the following: 1) if Asp to Glu mutants fail to rescue the growth defect as seen for the Asp to Ala mutants (KKT2 (D622A) and KKT3 (D692A)), and 2) assess if D to A and D to E mutations affect the integrity of the CL domain by comparing the SEC elution profiles of the recombinant mutants with corresponding profiles of the WT central domain (or by performing CD or 1D NMR experiments).

The authors suggest that the proteins co-purified by KKT2 (672-1030) are likely to be recruited to kinetochore via transient interactions involving KKT2. However, KKT2 RNAi only affected the localisation of KKT14 and KKT14 did not co-purify with KKT2 (672-1030). I think it would help if authors could comment on this in the manuscript.

The central domains of KKT2 and KKT3 appear to copurify with each other. Have the authors tested if central domains of KKT2 and KKT3 can interact with each other and if yes, does this contribute to the centromere localisation of KKT2 and KKT3?

In the case KKT2, the localisation data shown in Fig 3 show that the central domain, DPB and the linker region that connects the central and DPB domains are capable of associating with the centromere. Whereas, the DPB of KKT3 does not associate with centromeres and the centromere association seems to be exclusively mediated via the central domain. How similar are the KKT2 and KKT3 DPB domains? Are they expected to bind phospho-peptides as canonical polo-box domains? What could be the potential reason(s) for the strikingly different behaviour of the KKT2 and KKT3 DPB domains?

Minor points:

The authors should include percentage of sequence identity wherever sequence similarity is mentioned in the text.

Fig 7. Close up view of Zn coordination highlighting the involved Cys residues (like the one shown in Fig 6) will be useful.

Discussion section, line 7 states KKT4 localisation depends on KKT2. I think the authors refer to KKT14 here not KKT4?

We are grateful to the editor and reviewers for their comments and suggestions. We have addressed them as follows.

Editor's summary:

*You will see that the reviewers are supportive of the work but raise a number of points that we believe are important to address. In particular, there needs to be greater clarity on the relationship between the localization and DNA binding activity of the CL domain. While the *T. brucei* proteins did not work in the polarization-based assay, their DNA-binding activity should be tested using alternative approaches, such as gel shift/EMSA. Such an effort would help relate mutations generated in the central domain and their effects on localization to potential DNA binding activity.*

Response: We thank the editor for summarizing reviewers' comments. As suggested, we performed EMSA assays for the central domain of *TbKKT2*. We found that the wild-type protein bound the 50-bp centromere DNA probe and random DNA of different GC content with similar affinity, suggesting that the *TbKKT2* central domain has weak DNA-binding activity (micro molar range) and that it does not have a sequence specificity. We also tested a CL domain mutant (C597A, C600A) of the *TbKKT2* central domain and found that it retains DNA-binding activity in our EMSA assay. Because this mutant failed to localize at kinetochores in our ectopic expression experiments, these results suggest that DNA binding is not the mechanism for how the *TbKKT2* CL domain localizes at kinetochores. We have presented these results in Figure S4.

Reviewers also request clarification on a number of issues, additional controls and have important suggestions on points to cover in the text, all of which should be addressed in the revision.

Response: Please see our response to each comment below.

Finally, Reviewer #1 asks whether the kinase activities of KKT2 & KKT3 play a role in their kinetochore localization/functions and Reviewer #3 asks whether the Polo box domain resembles canonical phospho-peptide binding Polo box domains. Addressing these points would enhance the impact of the work, and we would encourage you to consider adding them into the revision if possible. However, we will not require these to be experimentally addressed but would recommend at the least that they be discussed in the text, as these points are very likely to occur to an interested reader based on the primary structures of KKT2 and KKT3.

Response: We are interested in understanding the function of the KKT2/3 kinase domains and have initiated their characterization *in vitro* and *in vivo*. However, because their thorough characterization will require a substantial amount of additional work, we would like to report it in another manuscript in the future. As for the polo boxes of KKT2/3, our previous sequence analysis showed that those residues in the polo boxes of human PLK1 that play key roles in phospho-peptide binding are not conserved in the divergent polo boxes of KKT2 and KKT3 (Figure 3 in Nerusheva and Akiyoshi, *Open Biology* 2016). It therefore remains unclear whether KKT2/3's divergent polo boxes are phosphorylation-dependent protein-protein interaction domains. It will be important to identify their direct interaction partners and to identify the mode of interactions. We have discussed these points in Discussion (line 335–343).

Reviewer #1 (Comments to the Authors (Required)):

Specific points:

1) *Figure 1: In Figure 1C, KKT3-YFP seems to be significantly reduced after KKT8 and KKT22 are depleted, which may not be consistent with their conclusion in text. Quantitative data should be shown to compare KKT2-YFP and KKT3-YFP localization with control vs various RNAi (e.g. percentage of cells showing YFP signals at kinetochores).*

Response: Although we agree that the KKT3-YFP signals in some of these images look reduced compared to controls, this is most likely because kinetochores are more scattered in these particular cells (i.e. not aligned at metaphase plate). In trypanosomes, there is a clustering of kinetochores or centromeres via an unknown mechanism, which causes the number of visible kinetochore dots to decrease as cells progress from prometaphase to metaphase. We had quantitative data in the legend in the original manuscript (“Dot formation was observed in >97% of 2K1N cells in all cases (n > 50, each)”), showing that kinetochore localization of KKT2 and KKT3 remained unaffected upon depletion of various kinetochore proteins.

2) *Figure 2B, E, H: I wonder what fraction of 2K1N cells show defective localization of YFP signals with control RNAi. Is it always 0%? If so, please clarify it in the figure legend. If not, results with control RNAi must be included in figures.*

Response: In the original manuscript, we had explained in the legend that “In each case, normal kinetochore signal was confirmed in un-induced 2K1N cells” without giving specific numbers. In the revised manuscript, we have included the following information into the legend: “In each case, defective localization of YFP signal was found in less than 6% (Figure 2B), 3% (Figure 2E) or 6% (Figure 2H) of un-induced 2K1N cells”. Given that the percentage of cells with defective localization was low in these controls, we did not include these results in figures for the sake of simplicity.

3) *Figure 7D and E: These results show that C646A C649A mutations do not change DNA binding affinity of PkKKT2 in vitro. However, corresponding mutations C616A and C619A abolish the centromere/kinetochore localization of TbKKT2 in vivo (Figure 8B). I wonder how the authors explain this discrepancy.*

Response: Together with our new data on the *TbKKT2* central domain that its DNA-binding activity was not abolished by CL domain mutations, we mentioned a possibility that DNA binding is unlikely to be the underlying mechanism for how the KKT2 CL domain functions in Discussion (line 355–357).

Optional point:

KKT2 and KKT3 are unique kinases that constitutively localize at centromeres/kinetochores and promote assembly of several other kinetochore proteins. I wonder if it is possible to address whether their kinase activity is required for such functions. For example, to address this, they can test whether kinase-negative forms of KKT2 and KKT3 rescue the defects caused by depletion of original KKT2 and KKT3. I suggest this experiment as optional, in case it is technically difficult or it requires considerable amount of time to complete.

Response: As we mentioned in our response to Editor’s comments, we have started characterizing the role of KKT2/3 kinase activities. However, because their full characterization will require substantial additional work, we will try to report it in another manuscript in the future.

Reviewer #2 (Comments to the Authors (Required)):

The results showing unaffected KKT2 and KKT3 localization in cells depleted for other KKT proteins is convincing. If I understand correctly, while they could only achieve efficient depletion of the kinetochore protein of interest in about 60% of cells (leaving 40% of cells in which depletion is incomplete), KKT2 and KKT3 signal was not observed in more than 94 or 97% of cells.

Response: It is true that efficient depletion of KKT proteins in RNAi-induced cells was limited to around 60%, meaning that 40% of cells still had signals (which is a limitation to our RNAi-based knockdown approach for these kinetochore proteins). However, KKT2 and KKT3 signal “was” observed in more than 94 or 97% of cells (which is a lot higher than 40%), suggesting that kinetochore localization of KKT2 and KKT3 was not affected by depleting these KKT proteins tested in Figure 1.

The data on the effect of KKT2 and 3 depletion on the recruitment of other kinetochore components are however weaker, probably due to technical reasons related to inefficient depletions of these proteins (as also stated in the text and the figure legend). For Figure 2H, data of only a subset of kinetochore components is displayed. To enable a better comparison with the single depletions, it would be good if the same kinetochore components as those shown in B and E would be analyzed in this panel as well.

Response: We performed KKT2/3 double depletion experiments for YFP-KKT14 and found that KKT14 localization was defective in 22% of cells, which is lower than 46% found in the KKT2 single RNAi. These results suggest that depletion of KKT2 in KKT2/3 double RNAi is not as efficient as that in KKT2 single RNAi. We added this result into Figure 2H. This result is consistent with the observation that there were residual KKT2 signals in Figure 2I. Due to this limitation, we did not attempt to examine the localization of other affected kinetochore proteins in the KKT2/3 double RNAi. It will be important to establish a better knockdown method for these proteins in the future.

In general, it would also be helpful if the authors specify how they determined absence of a kinetochore protein. Was a certain threshold of intensity used to conclude presence or absence? If data on signal intensities are available, I think it would contribute to the study if those were displayed to discuss potential quantitative effects on kinetochore protein localization.

Response: It has been extremely difficult to obtain reliable quantitative data on the signal intensity of kinetochore dots in trypanosomes for various reasons. For example, in trypanosomes, there is an apparent clustering of kinetochores or centromeres in mitosis (via an unknown mechanism), which causes the number of visible kinetochore dots to decrease as cells progress from G2 to metaphase (2K1N cells). This clustering makes the brightness of kinetochore dots highly variable even in wild-type G2/prometaphase/metaphase cells. Furthermore, cells have alignment problems after RNAi of essential KKT genes, which means that we were unable to choose metaphase-like cells in most cases. We are currently trying to establish a machine learning-based automated image quantification method, but it still has several major hurdles to overcome before it can provide reliable and meaningful data. For these reasons, we had to count the number of cells that clearly had or did not have detectable kinetochore dots manually in this study. We added this explanation in the methods (line 432–434).

There is a bit of a discrepancy between the identity of kinetochore components with affected localization upon KKT2 or KKT3 depletion and those found in their MS data. Of course, kinetochore components with affected centromere localization might not be direct protein binding partners of KKT2 or KKT3 and thus not identified in their proteomic datasets. One could have however expected that the localization of protein binding partners would be

affected upon KKT2 or KKT3 depletions. If the authors agree, I think it would be good if they comment on this aspect.

Response: It is important to note that immunoprecipitation experiments do not necessarily reveal direct interaction partners. In fact, we have not yet identified any direct interaction partner for KKT2 or KKT3. Because our immunoprecipitation and mass spectrometry experiments showed some overlapping results in co-purifying kinetochore proteins (e.g. KKT1 and KKT7), we think that the lack of protein mis-localization for co-purifying proteins in KKT2 or KKT3 individual RNAi experiments is likely due to redundancy (e.g. KKT1) and/or inefficient depletion in the case of KKT2/3 double RNAi experiments (e.g. KKT7).

Proteins identified in KKT2(558-679) IP-MS: Are peptides in Table of panel C mapping to the full-length or only the expressed fragment of KKT2? If the authors recovered the full-length KKT2 protein, could the recruitment of the fragment also be due to dimerization of the protein in this part instead of DNA binding? Same for KKT3, Figure 4E.

Response: We detected only negligible levels of KKT2 peptides corresponding to outside of 558-679, suggesting that KKT2 (558-679) does not co-purify with full-length KKT2 at least in this assay condition. We obtained similar results for the KKT3 central domain.

Figure 8. In contrast to KKT3-CL mutants that interestingly are completely unable to localize to centromeres, KKT2 CL mutants can at least at time 0 prior to RNAi induction against the endogenous copy, probably due to the interaction with other kinetochore proteins as hypothesized by the authors. It would be interesting to test whether KKT2 CL mutant localization is impaired at later time points when the growth defects arise. This would suggest that kinetochore complex might indeed be destabilized in the presence of these mutations.

Response: We have looked at the localization of KKT2 CL mutants at 96 hr post-induction and found that they still formed kinetochore-like dots (100%, N > 50 cells), suggesting that the CL-independent kinetochore localization mechanisms are sufficient for maintaining KKT2 at kinetochores even in the absence of endogenous KKT2 proteins. We have added this result in Figure S5D.

A model for kinetochore assembly summarizing the findings of this paper and the current knowledge would be very helpful include at the end of the paper.

Response: We have made Figure 10 that summarizes the findings of this paper and the current knowledge of the kinetoplastid kinetochore assembly.

I think there is a mix-up between the links for the supplemental tables. Link to Table S1 opens Table S3 for example...

Response: Thanks for pointing out the issue. We have fixed it in the revised manuscript.

Reviewer #3 (Comments to the Authors (Required)):

How the central domains of KKT2 and KKT3 contribute to their centromere localisation is still not clear. The authors propose that the central domain of KKT2 and KKT3 might have DNA-binding activity. Due to technical difficulty the authors could not test this hypothesis for KKT2 and KKT3, but instead they show that Pk KKT2a CL-like domain can indeed bind DNA. In my opinion, testing the DNA binding ability of the central domains of KKT2 and KKT3 by EMSA (Electrophoretic Mobility Shift Assay) or ITC (Isothermal Titration Calorimetry) will improve the impact of this work and needs to be tested.

Response: We thank the reviewer for this comment. As we mentioned above, we performed EMSA assays for the central domain of *TbKKT2*. We found that the wild-type protein bound 50-bp centromere DNA probe and 50-bp random DNA of different GC content (26% and 74%) with similar affinities, suggesting that, similarly to *PkKKT2*, the *TbKKT2* central domain has weak DNA-binding activity (micro molar range) and that it does not have a sequence specificity. We also tested a CL domain mutant (C597A, C600A) of the *TbKKT2* central domain that had failed to localize at kinetochores in our ectopic expression experiments. We found that this mutant retains DNA-binding activity in our EMSA assay, suggesting that kinetochore localization of the *TbKKT2* CL domain is not dependent on its DNA-binding activity. We have added these results in Figure S4. Unfortunately, we could not test DNA-binding activity for *TbKKT3* because we were unable to purify this construct despite multiple attempts.

Surprisingly, disrupting the integrity of the PkKKT2a CL domain by mutating the Zn coordinating cys residues did not affect its ability to bind DNA. The authors do not really comment about this intriguing observation. This clarification is important as equivalent mutations in Tb KKT2 and KKT3 (disrupting the integrity of CL domain) affect their centromere localisation.

Response: As suggested, we added a comment that the main function of the CL domain is unlikely to be DNA binding in Discussion (line 355–357).

Mutating the conserved Asp residue of the negatively charged surface (in KKT2 CL and KKT3 CL) affects centromere localisation of KKT2 and KKT3. What I find intriguing is that a subtle variation at this position, ASP to GLU (which is unlikely to change the electrostatic surface charge of the patch), is sufficient to disrupt the centromere localisation. I would like to see the authors explaining the rationale for mutating this residue to GLU and comment on the observation.

Response: We thank the reviewer for this comment. We made Asp to Glu mutations based on our finding that the Asp residue (D622 in *TbKKT2*, D692 in *TbKKT3*; Figure S3) is strictly conserved in all kinetoplastids. It is indeed intriguing that such a subtle change affected the localization of the *KKT2* central domain fragment or full-length *KKT3* protein. We added a comment on this observation in the manuscript (line 284). We expect that further understanding of the function of the *KKT2/3* central domains will provide insights into why such a subtle change disrupted their kinetochore localization.

It would certainly help if authors could test the following: 1) if Asp to Glu mutants fail to rescue the growth defect as seen for the Asp to Ala mutants (KKT2 (D622A) and KKT3 (D692A)), and 2) assess if D to A and D to E mutations affect the integrity of the CL domain by comparing the SEC elution profiles of the recombinant mutants with corresponding profiles of the WT central domain (or by performing CD or 1D NMR experiments).

Response: We agree that further characterization of Asp to Glu mutants *in vivo* and *in vitro* would be important for better understanding of this invariant Asp residue. However, because we already showed its importance by the D to A mutation and because significant amounts of additional work will be required to address this comment, we feel that it is beyond the scope of this manuscript.

The authors suggest that the proteins co-purified by KKT2 (672-1030) are likely to be recruited to kinetochore via transient interactions involving KKT2. However, KKT2 RNAi only affected the localisation of KKT14 and KKT14 did not co-purify with KKT2 (672-1030). I think it would help if authors could comment on this in the manuscript.

Response: It is indeed interesting that *KKT2* RNAi only affected the localization of *KKT14*, although *KKT14* did not co-purify with *KKT2* fragments we tested. However, our previous

study showed that KKT2 was one of the most abundant proteins in the immunoprecipitates of KKT14 (Akiyoshi and Gull, Cell 2014). It is therefore possible that KKT14 co-purifies with a KKT2 fragment that we did not test. We have added these comments (line 147–151).

The central domains of KKT2 and KKT3 appear to copurify with each other. Have the authors tested if central domains of KKT2 and KKT3 can interact with each other and if yes, does this contribute to the centromere localisation of KKT2 and KKT3?

Response: We have not been able to test this possibility because purification of the *Tb*KKT3 central domain has been unsuccessful. Although kinetochore localization of KKT2 and KKT3 appeared independent based on our RNAi data, we agree with the reviewer that it will be important to dissect the relationship between KKT2 and KKT3.

In the case KKT2, the localisation data shown in Fig 3 show that the central domain, DPB and the linker region that connects the central and DPB domains are capable of associating with the centromere. Whereas, the DPB of KKT3 does not associate with centromeres and the centromere association seems to be exclusively mediated via the central domain. How similar are the KKT2 and KKT3 DPB domains? Are they expected to bind phospho-peptides as canonical polo-box domains? What could be the potential reason(s) for the strikingly different behaviour of the KKT2 and KKT3 DPB domains?

Response: KKT2 DPB and KKT3 DPB are 25% identical in protein sequences (Akiyoshi and Gull, Cell, 2014). As we mentioned above, our previous sequence analysis showed that those residues in polo boxes of human PLK1 that play key roles in phospho-peptide binding are not conserved in the divergent polo boxes of KKT2 and KKT3 (Figure 3 in Nerusheva and Akiyoshi, Open Biology 2016). It therefore remains unclear whether KKT2/3's divergent polo boxes are phosphorylation-dependent protein-protein interaction domains. It will be important to identify their direct interaction partners and to reveal the mode of interactions. We have shown that the KKT3 DPB fragment can localize at kinetochores in anaphase. However, we also showed that mutations in the central domain abolished the localization of the full-length KKT3 protein. Our interpretation is that affinity of KKT3 DPB to other kinetochore proteins is not as high as that of KKT2 DPB. Together with our finding that the KKT3 central domain fragment localizes at kinetochores more robustly than the KKT2 central domain, we speculate that KKT3 is more specialized for centromere localization and that KKT2 is more specialized for interacting/recruiting other kinetochore proteins.

Minor points:

The authors should include percentage of sequence identity wherever sequence similarity is mentioned in the text.

Response: As suggested, we have added percentage of sequence identity where we mentioned sequence similarity (lines 156, 215, 216, 218, 231, 300, and 349).

Fig 7. Close up view of Zn coordination highlighting the involved Cys residues (like the one shown in Fig 6) will be useful.

Response: We thank the reviewer for the comment. We added the close-up view to Figure 7 in the revised manuscript.

Discussion section, line 7 states KKT4 localisation depends on KKT2. I think the authors refer to KKT14 here not KKT4?

Response: We do refer to KKT4: KKT4 is important for recruiting "KKT20" (Llauro, JCB 2018).

May 10, 2021

RE: JCB Manuscript #202101022R

Dr. Bungo Akiyoshi
University of Oxford
South Parks Road
Oxford OX1 3QU
United Kingdom

Dear Dr. Akiyoshi,

Thank you for submitting your revised manuscript entitled "Unconventional kinetochore kinases KKT2 and KKT3 have unique centromere localization domains." We would be happy to publish your paper in JCB pending final revisions necessary to meet our formatting guidelines (see details below).

A. MANUSCRIPT ORGANIZATION AND FORMATTING:

Full guidelines are available on our Instructions for Authors page, <https://jcb.rupress.org/submission-guidelines#revised>. **Submission of a paper that does not conform to JCB guidelines will delay the acceptance of your manuscript.**

- 1) Text limits: Character count for Articles is < 40,000, not including spaces. Count includes title page, abstract, introduction, results, discussion, and acknowledgments. Count does not include materials and methods, figure legends, references, tables, or supplemental legends.
- 2) Figures limits: Articles may have up to 10 main text figures.
- 3) Figure formatting: Scale bars must be present on all microscopy images, including inset magnifications. Molecular weight or nucleic acid size markers must be included on all gel electrophoresis.
- 4) Statistical analysis: Error bars on graphic representations of numerical data must be clearly described in the figure legend. The number of independent data points (n) represented in a graph must be indicated in the legend. Statistical methods should be explained in full in the materials and methods. For figures presenting pooled data the statistical measure should be defined in the figure legends. Please also be sure to indicate the statistical tests used in each of your experiments (both in the figure legend itself and in a separate methods section) as well as the parameters of the test (for example, if you ran a t-test, please indicate if it was one- or two-sided, etc.). Also, if you used parametric tests, please indicate if the data distribution was tested for normality (and if so, how). If not, you must state something to the effect that "Data distribution was assumed to be normal but this was not formally tested."
- 5) Title: We feel that the title should mention that the work is in kinetoplastids and not include

'kinase' since the functions of the kinase domains were not addressed in the study. Therefore we suggest the following title: "Kinetoplastid kinetochore proteins KKT2 and KKT3 have unique centromere localization domains."

6) Materials and methods: Should be comprehensive and not simply reference a previous publication for details on how an experiment was performed. Please provide full descriptions (at least in brief) in the text for readers who may not have access to referenced manuscripts. The text should not refer to methods "...as previously described."

7) Please be sure to provide the sequences for all of your primers/oligos and RNAi constructs in the materials and methods. You must also indicate in the methods the source, species, and catalog numbers (where appropriate) for all of your antibodies.

8) Microscope image acquisition: The following information must be provided about the acquisition and processing of images:

a. Make and model of microscope

b. Type, magnification, and numerical aperture of the objective lenses

c. Temperature

d. Imaging medium

e. Fluorochromes

f. Camera make and model

g. Acquisition software

h. Any software used for image processing subsequent to data acquisition. Please include details and types of operations involved (e.g., type of deconvolution, 3D reconstitutions, surface or volume rendering, gamma adjustments, etc.).

10) Supplemental materials: There are strict limits on the allowable amount of supplemental data. Articles/Tools may have up to 5 supplemental figures and 10 videos. Please also note that tables, like figures, should be provided as individual, editable files. A paragraph summary of all supplemental material should appear at the end of the Materials and methods section.

11) eTOC summary: A ~40-50 word summary that describes the context and significance of the findings for a general readership should be included on the title page. The statement should be written in the present tense and refer to the work in the third person. It should begin with "First author name(s) et al..." to match our preferred style.

13) A separate author contribution section is required following the Acknowledgments in all research manuscripts. All authors should be mentioned and designated by their first and middle initials and full surnames. We encourage use of the CRediT nomenclature (<https://casrai.org/credit/>).

14) ORCID IDs: ORCID IDs are unique identifiers allowing researchers to create a record of their various scholarly contributions in a single place. At resubmission of your final files, please consider providing an ORCID ID for as many contributing authors as possible.

B. FINAL FILES:

Thank you for this interesting contribution, we look forward to publishing your paper in Journal of Cell Biology.

Sincerely,

Arshad Desai, PhD
Monitoring Editor
Journal of Cell Biology

Dan Simon, PhD
Scientific Editor
Journal of Cell Biology

Reviewer #1 (Comments to the Authors (Required)):

The authors have satisfactorily addressed all my specific points by conducting new experiments and by adding more information in text. The EMSA is indeed a good addition to the paper. I recommend the revised manuscript for publication in JCB.